# Lacking mechanistic disease definitions and corresponding association data hamper progress in network medicine and beyond

Sepideh Sadegh [1,2], James Skelton[3], Elisa Anastasi[3], Andreas Maier [2], Klaudia Adamowicz[2], Anna Möller [4], Nils M. Kriege [5,6], Jaanika Kronberg [7], Toomas Haller[7], Tim Kacprowski [8,9], Anil Wipat[3,11], Jan Baumbach [2,10,11] & David B. Blumenthal [4,11] ✉

A long-term objective of network medicine is to replace our current, mainly phenotype-based disease definitions by subtypes of health conditions corresponding to distinct pathomechanisms. For this, molecular and health data are modeled as networks and are mined for pathomechanisms. However, many such studies rely on large-scale disease association data where diseases are annotated using the very phenotype-based disease definitions the network medicine field aims to overcome. This raises the question to which extent the biases mechanistically inadequate disease annotations introduce in disease association data distort the results of studies which use such data for pathomechanism mining. We address this question using global- and local-scale analyses of networks constructed from disease association data of various types. Our results indicate that large-scale disease association data should be used with care for pathomechanism mining and that analyses of such data should be accompanied by close-up analyses of molecular data for well-characterized patient cohorts.

Since the seminal articles by Goh et al. [1] and Barabási et al. [2], network medicine has developed into an increasingly mature and diverse research field with its own dedicated journals[3], associations[4], and subfields. One of the network medicine field's long-term objectives is to replace our current mainly phenotype-based disease classification systems by a mechanistically grounded disease vocabulary[5–7]. In such a vocabulary, phenotype-based disease definitions are replaced by so-called endotypes, i.e., distinct molecular mechanisms underlying the disease phenotypes. Once properly disentangled into disjoint,

individually targetable endotypes[5], disease-modifying treatment strategies might become available for diseases which, at the moment, can be treated only symptomatically.

Two clarifications are required to define the scope of this paper: Firstly, we use the term "endotype" to denote molecular endotypes as explained by Anderson[8], Lötvall et al. [9], and Nogales et al. [5] – i.e., the underlying molecular mechanisms driving disease phenotypes. There are other works where the term "endo(patho)phenotype" denotes common intermediate phenotypes[6] such as inflammation, fibrosis, or

[1]Chair of Experimental Bioinformatics, TUM School of Life Sciences, Technical University of Munich, Munich, Germany. [2]Institute for Computational Systems Biology, University of Hamburg, Hamburg, Germany. [3]School of Computing, Newcastle University, Newcastle upon Tyne, UK. [4]Biomedical Network Science Lab, Department Artificial Intelligence in Biomedical Engineering, Friedrich-Alexander-Universität Erlangen-Nürnberg, Erlangen, Germany. [5]Faculty of Computer Science, University of Vienna, Vienna, Austria. [6]Research Network Data Science, University of Vienna, Vienna, Austria. [7]Estonian Genome Centre, Institute of Genomics, University of Tartu, Tartu, Estonia. [8]Division Data Science in Biomedicine, Peter L. Reichertz Institute for Medical Informatics of Technische Universität Braunschweig and Hannover Medical School, Braunschweig, Germany. [9]Braunschweig Integrated Centre of Systems Biology (BRICS), TU Braunschweig, Braunschweig, Germany. [10]Computational Biomedicine Lab, Department of Mathematics and Computer Science, University of Southern Denmark, Odense, Denmark. [11]These authors jointly supervised this work: Anil Wipat, Jan Baumbach, David B. Blumenthal. ✉e-mail: david.b.blumenthal@fau.de

thrombosis which drive phenotypic disease manifestations[10,11]. Secondly, we would like to stress that compiling a endotype-based disease vocabulary is a genuinely biomedical rather than a semantic endeavor: It does not consist in redefining semantic relationships between existing disease terms but in uncovering currently unknown molecular disease mechanisms and dissecting umbrella diseases such as Alzheimer's disease or coronary artery disease into endotypes which are clearly characterized at a molecular level[5].

In order to reach the objective of an endotype-based disease vocabulary, network medicine approaches aim at uncovering pathomechanisms driving diseases. Here, we broadly distinguish between close-up and bird's-eye-view (BEV) network medicine approaches, depending on the data used as primary input towards this task (this distinction is of course an idealized binarization of a continuous spectrum, but serves as a conceptual framework for this article). Close-up network medicine studies focus on a specific disease and start their analyses with molecular data for well-characterized patient cohorts. Such studies are typically carried out as close collaborations between bioinformaticians and domain experts from the biomedical sciences. They tend to be time- and labor-intensive and often involve the development or customization of data analysis methods for specific datasets. The most impressive translational results of the network medicine field have been reached via such close-up studies. For instance, close-up studies have led to novel mechanistic insights into type 2 diabetes[12], liver fibrosis[13], pulmonary arterial hypertension[14], asthma[15], hypertrophic cardiomyopathy[16], pre-eclampsia[17], chronic obstructive pulmonary disease, and idiopathic pulmonary fibrosis[18].

In contrast to that, BEV approaches use large-scale disease association data that are typically gathered from several data sources. Various studies have generated evidence for the validity of this overall approach: For instance, Menche et al. [19] demonstrated that disease-associated genes form so-called disease modules, i.e., highly connected subnetworks within protein-protein interaction (PPI) networks, and that biological and clinical similarity of two diseases results in significant topological proximity of these modules. In a similar vein, Iida et al. [20] showed that shared therapeutic targets or shared drug indications are correlated with high topological module proximity. Guney et al. [21] and Cheng et al. [22] showed that the network-based separation between drug targets and disease modules is indicative of drug efficacy. Cheng et al. [23] and Zhou et al. [24] found that FDA-approved drug combinations are proximal to each other and to the modules of the targeted diseases in the interactome.

Despite the promising findings summarized above, several studies have pointed out important biases in the data used by BEV approaches. Menche et al. [19] have studied the effect of incompleteness of disease-gene association and protein-protein interaction (PPI) data on network medicine. Schaefer et al. [25] have shown that the previously observed[26–28] high node degree of cancer-associated proteins in PPI networks can largely be explained by the fact that cancer-associated proteins are tested more often for interaction than others. Lazareva et al. [29] found that widely used methods to mine PPI networks for pathomechanisms inherit this bias in that they mainly learn from the node degrees instead of exploiting the biological knowledge encoded in the edges of the PPI networks. Haynes et al. [30] showed that study bias also distorts functional gene annotation resources such as the Gene Ontology (GO)[31]. Kustacher et al. [32] made a similar point for functional protein annotations and sketched a roadmap for systematically exploring the understudied part of the proteome. Stoeger et al. [33] and Rodriguez-Esteban[34] looked into reasons that might lead to the emergence of gene study bias and identified, respectively, a limited number of biological characteristics[33] and speed of information propagation between scientific communities as potential drivers[34].

While the aforementioned studies have analyzed the impact of various types of data biases related to genes and proteins (and, to a lesser extent, also variants), the disease part of disease-gene and other disease association data introduces another, so far unstudied type of data bias: In currently available large-scale disease association data, diseases are annotated with the very phenotype-based disease definitions the network medicine field aims to overcome. BEV approaches hence risk to systematically reproduce the biases introduced by these disease definitions. Consequently, BEV approaches make the implicit assumption that the biases introduced by phenotype-based disease definitions even out and that, despite those biases, disease association data using these definitions still contain useful information about the pathomechanism that are to be uncovered.

In this work, we quantify to which extent this implicit assumption is indeed backed by data. Towards this end, we construct disease-disease networks (called "diseasomes" in the remainder of this article) based on (1) disease-gene associations, (2) disease-variant associations, (3) comorbidity data, (4) symptom data, and (5) drug-indication data, as well as drug-disease and drug-drug networks (called "drugomes") based on drug-indication and drug-target data. We then formulate two testable hypotheses that follow from the implicit assumption of BEV network medicine: The global-scale hypothesis states that, globally, networks constructed from two different types of association data that both contain useful information about endotypes should be pairwise more similar than expected by chance. The local-scale hypothesis states that this should hold not only globally but also for the neighborhoods of the individual diseases and drugs represented by nodes in the constructed networks.

In line with the findings of prior studies[20–24], our analyses provide solid evidence for the global-scale hypothesis. However, they only partially support the local-scale hypothesis. Figuratively speaking, BEV network medicine hence only allows a distal view at the endotypes that are to be discovered. When zooming in on individual diseases, the picture becomes blurred and less reliable (see Fig. 1 for a conceptual visualization and Fig. 2 for a concrete exemplification of this phenomenon in the context of neurodegenerative diseases). This implies that, in order to yield translational results, BEV approaches need to be supplemented with additional layers of molecular data for well-characterized patient cohorts and a dedicated focus on the specific diseases which are being investigated. In particular, fine-grained molecular patient data are crucial for implementing network medicine's long-term objective to replace current phenotype- or organ-based disease definitions by mechanistically grounded endotypes. The main finding of this study is hence that the biases current disease definitions introduce in large-scale disease association databases such as OMIM and DisGeNET do not even out and that such databases should be used with care in all fields of data-centric biomedicine: Instead of blindly using public disease association data out of convenience for pathomechanism mining, we strongly recommend biomedical researchers to always consciously ponder to which extent biases in these data introduced by phenotype-based disease terms threaten to distort their potential findings.

## Results
### Neurodegenerative diseases as case example
Before presenting the comprehensive results of our analyses, we visualize the phenomenon of local blurriness in BEV network medicine with a small example. We compiled a list of diseases that fall under the parent term "neurodegenerative disease" in the MONDO disease hierarchy. From those, we kept diseases for which we have nodes in the aligned gene- and drug-based diseasomes. This led to a cluster of seven neurodegenerative diseases which are highly connected in both diseasomes. Figure 2 shows this cluster, together with the contained diseases' local empirical P-values obtained from the comparison of gene- and drug-based diseasomes in MONDO space, the global empirical P-value, as well as the cluster-level empirical P-value (see next subsection and Methods for explanations on how we obtained the P-values). While only two local empirical P-values are significant at 0.05

level, the cluster-level and global empirical *P*-values are significant at levels 0.01 and 0.001, respectively.

## Overview of analyses

Let *D* be disease association data of some data type *T* commonly used by BEV approaches (e.g., disease-gene associations). Further assume that *D* contains entries $D(d_1)$ and $D(d_2)$ for two diseases $d_1$ and $d_2$ that share an unknown molecular disease mechanism. Then this shared mechanism should lead to similarities between $D(d_1)$ and $D(d_2)$, given that *D* indeed contains useful information about disease mechanisms[35]. For instance, we would expect that the diseases $d_1$ and $d_2$ have similar profiles of disease-associated genes, that they exhibit high comorbidity, that they lead to similar symptoms, and that they can be treated by similar drugs. We can capture such similarities in

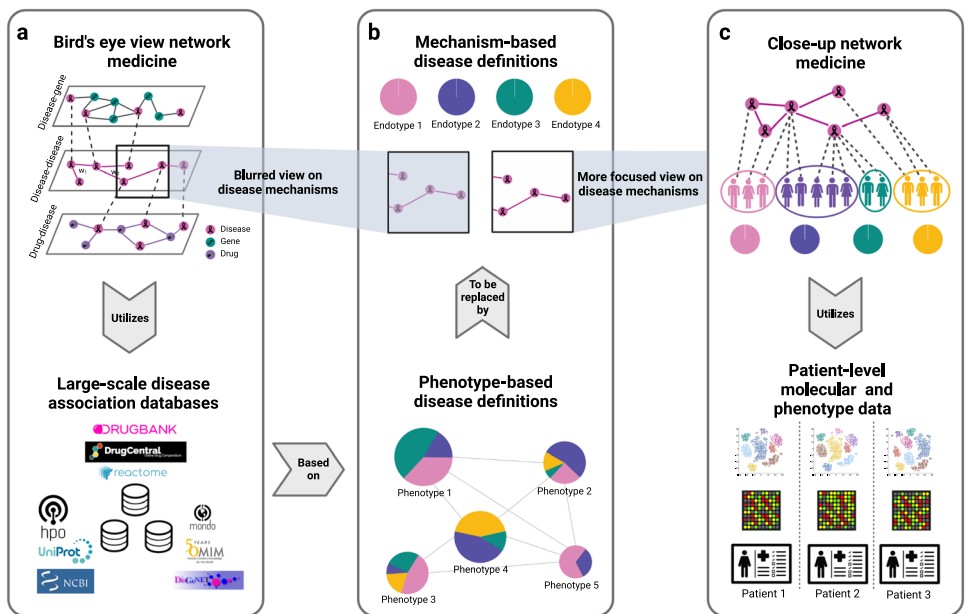

**Fig. 1 | BEV vs. close-up network medicine. a** BEV network medicine mainly utilizes large-scale disease association data where diseases are annotated with phenotype-based disease definitions (**b**, bottom). BEV network medicine inherits the bias introduced by these definitions, which leads to a blurred view on individual pathomechanisms (**b**, top). **c** Close-up network medicine uses patient-level molecular data and is hence less dependent on the phenotype-based disease definitions that network medicine aims to replace by mechanism-based endotypes.

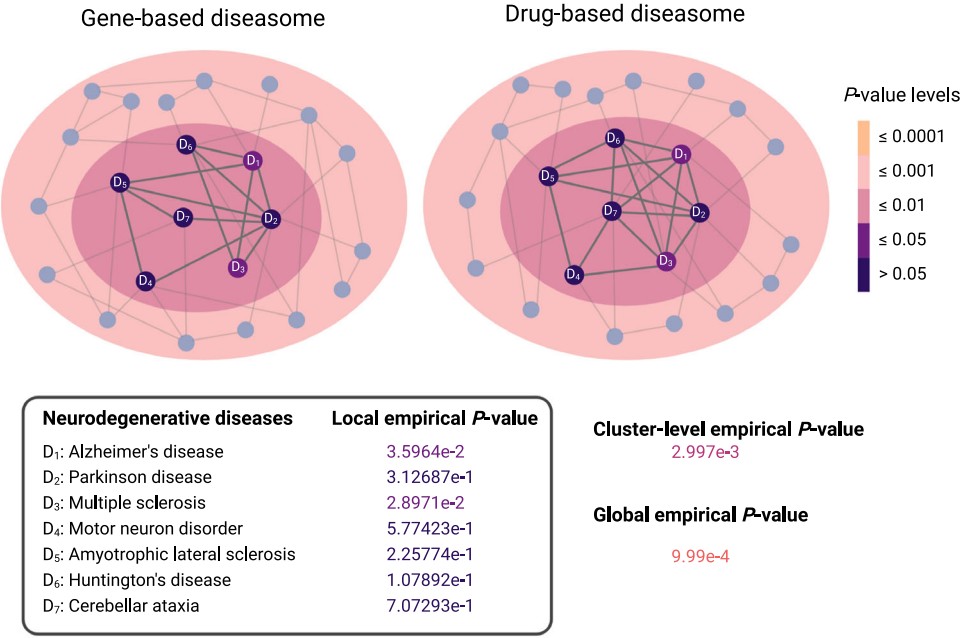

| Neurodegenerative diseases | Local empirical *P*-value |
|---|---|
| $D_1$: Alzheimer's disease | 3.5964e-2 |
| $D_2$: Parkinson disease | 3.12687e-1 |
| $D_3$: Multiple sclerosis | 2.8971e-2 |
| $D_4$: Motor neuron disorder | 5.77423e-1 |
| $D_5$: Amyotrophic lateral sclerosis | 2.25774e-1 |
| $D_6$: Huntington's disease | 1.07892e-1 |
| $D_7$: Cerebellar ataxia | 7.07293e-1 |

**Cluster-level empirical *P*-value**
2.997e-3

**Global empirical *P*-value**
9.99e-4

**Fig. 2 | Locally blurred results for neurodegenerative diseases.** The color gradient visualizes local-, global-, and cluster-level empirical *P*-values (one-sided, unadjusted) obtained from the comparison of gene- and drug-based diseasomes in MONDO vocabulary. The gene-based diseasome was constructed based on disease-gene association data integrated from DisGeNET[36] and OMIM[43] and two diseases were connected by an edge if they share at least one disease associated gene. The drug-based diseasome was constructed based on drug-indication data integrated from CTD[48] and DrugCentral[37] and two diseases were connected by an edge if they share at least one indicated drug.

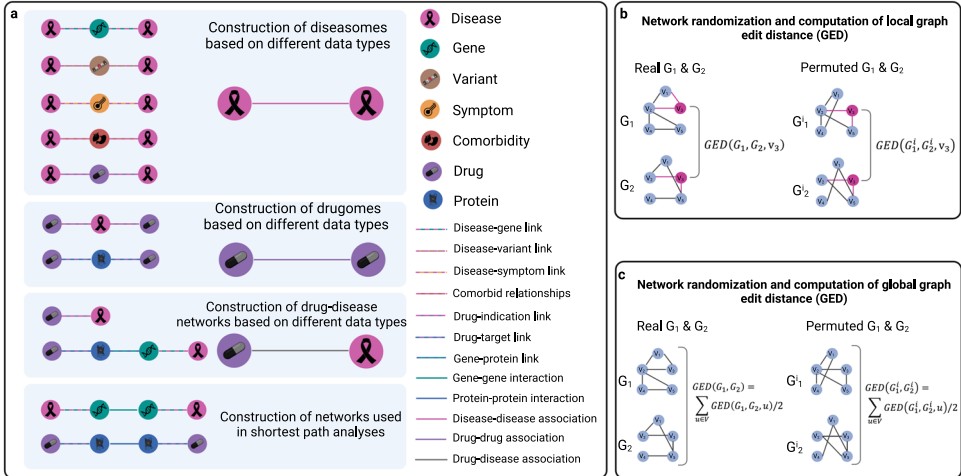

**Fig. 3 | Overview of compared networks and graph edit distance computation. a** We compared five different types of disease-disease networks (diseasomes), two different types of drug-drug networks (drugomes), and two different types of drug-disease networks. Pairwise comparisons between those networks were carried out using local and global graph edit distance (GED). **b** Local GED was used to quantify the dissimilarities of the individual nodes' neighborhoods across different networks in comparison to pairs of randomly rewired networks. **c** Global network dissimilarities were computed using global GED, obtained by summing up the local GEDs of the individual nodes.

diseasomes, where diseases $d_1$ and $d_2$ are connected by an edge if $D(d_1)$ and $D(d_2)$ are sufficiently similar. In order to assess the implicit assumption of BEV network medicine approaches with quantitative means, we hence formulate the following testable hypotheses (see Methods for an argument to support these hypotheses):

- Global-scale hypothesis: For all disease association data $D_1$ and $D_2$ that are assumed to contain useful information about endotypes (e.g., disease-gene association and drug-indication data from databases such as DisGeNET[36] and DrugCentral[37]), diseasomes $G_1$ and $G_2$ constructed based on $D_1$ and $D_2$ should be pairwise more similar than expected by chance.
- Local-scale hypothesis: For all disease association data $D_1$ and $D_2$ that are assumed to contain useful information about endotypes and any disease term $d$ that appear in $D_1$ and $D_2$, the direct neighborhood of $d$ in the diseasomes $G_1$ and $G_2$ constructed based on $D_1$ and $D_2$ should be pairwise more similar than expected by chance. For example, under the assumption that disease-gene and drug-indication databases such as DisGeNET and DrugCentral contain useful information about Alzheimer's disease (AD) mechanisms, there should be a significant overlap between the set of diseases whose associated genes overlap with AD-associated genes and the set of diseases which can be treated with drugs also indicated for AD.

To test these two hypotheses, we constructed various diseasomes, drugomes, and drug-disease networks based on different data types. An overview of the used data types and derived networks is shown in Fig. 3a. Using customized versions of the graph edit distance (GED)[38,39], we then compared these networks in a pairwise manner both on a local scale, i.e. zoomed-in on individual disease or drug nodes, and on a global scale. More precisely, we generated 1000 permuted networks as randomized counterparts for each network. Subsequently, we compared the distributions of local and global GEDs obtained for the original networks to GED distributions obtained for randomized counterparts. Network randomization and computation of local and global GED are illustrated in Fig. 3b, c. While local GED measures the dissimilarity between the individual nodes' neighborhoods in the compared networks, global GED is a measure for the overall dissimilarity of the networks.

We also evaluated how annotating the data using disease vocabularies of different granularity affect the results, by carrying out the

analyses using MONDO IDs[40] and UMLS CUIs[41] (finer granularity) and ICD-10[42] three-character codes (coarser granularity) as node IDs in the constructed networks, respectively. To this end, where possible, we constructed the networks in MONDO, UMLS CUI, and in ICD-10 vocabulary (using three-character level codes). Note that analyses involving comorbidity data were carried out only in ICD-10 and the comparison between target- and indication-based drugomes only in MONDO vocabulary (see Methods for an explanation). Moreover, neither the semantic layers of the MONDO disease ontology nor the hierarchy of the UMLS CUI and ICD-10 classification system were used to add edges to our diseasomes. MONDO, UMLS CUI, and ICD-10 were only used as vocabularies, i.e., to provide the node IDs in our networks. Whether two disease nodes are connected by an edge exclusively depends on the primary databases containing the association data (upon mapping to MONDO, UMLS CUI, or ICD-10). For instance, two diseases are connected in the gene-based diseasome in MONDO vocabulary if the intersection of the sets of genes associated with their MONDO IDs is non-empty, where disease-gene associations were obtained from OMIM[43] and DisGeNET[36].

GED quantifies the dissimilarity between two networks as the minimum cost of an edit path transforming one network into the other. Edit paths are sequences of elementary edit operations (node and edge insertions, substitutions, and deletions), all of which come with associated edit costs. Hence, the GED is a distance measure between two networks. We computed three different versions of GED using uniform, weight-based, and rank-based edge editing costs, respectively. Uniform edit costs discard the association strengths of the edges in the compared networks; weight- and rank-based edit costs incorporate them by making it more expensive to delete or insert edges with strong associations or to substitute them by edges with weak associations. Corroborating the robustness of our analysis method, we obtained similar results for all three versions of GED. In the following, only the results of uniform edit costs are reported. Results for rank- and weight-based edit costs can be found in Supplementary Figs. 1–4 and 9–12, respectively. More details on disease vocabulary mapping, network construction, and GED computation can be found in Methods.

## Results of global-scale analyses

To test the global-scale hypothesis, we computed empirical *P*-values for each pair of networks based on global GEDs (Fig. 4a, left panel). For all evaluated pairs of networks (in MONDO, UMLS CUI, and ICD-10

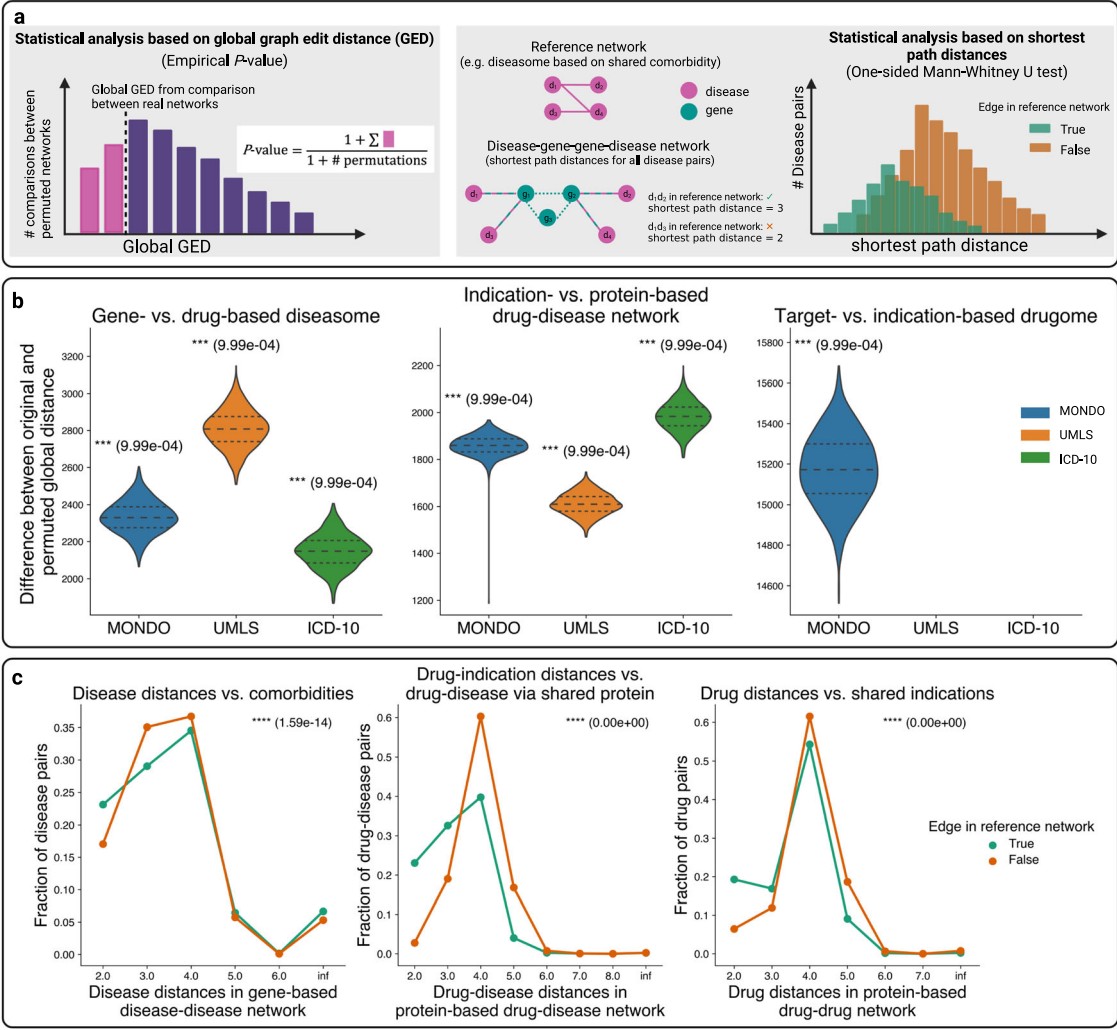

**Fig. 4 | Global-scale analyses. a** Illustration of global-scale analysis methods. Left panel: Statistical analyses based on global GED via empirical *P*-values. Right panel: Statistical analyses based on shortest path distances via MWU test. **b** Differences of global GEDs (based on uniform edge edit costs) between a selection of original networks and their counterpart permuted networks, and corresponding global empirical *P*-values (one-sided, unadjusted) in MONDO, UMLS, and ICD-10 vocabularies. All obtained global empirical *P*-values are at the lower resolution limit of our permutation tests with 1000 randomized network pairs. **c** Selected results of

shortest path analyses and the corresponding MWU *P*-values (one-sided, unadjusted). Left: Disease distances in gene-based disease-disease network vs. comorbidity-based diseasome as the reference network. Middle: Drug-disease distances in protein-based drug-disease network vs. drug-indication network as the reference network. Right: Drug distances in protein-based drug-drug network vs. indication-based drugome as the reference network. All networks underlying the results shown in (**c**) are constructed in the MONDO vocabulary.

vocabularies), we obtained smaller global GEDs for the original diseaseomes, drugomes, or drug-disease networks than for randomized counterparts, leading to empirical *P*-values which are significant at 0.001 level. Differences between GEDs obtained for permuted and a selection of original networks are shown in Fig. 4b. For the full results of our global-scale analyses, see Supplementary Fig. 5.

Moreover, we performed analyses based on shortest path distances between disease-disease, drug-drug, and drug-disease pairs in disease-gene-gene-disease, drug-protein-protein-drug, and disease-protein-protein-drug networks, where protein-protein and gene-gene links were obtained from PPIs. We then compared shortest path distances for node pairs which do and node pairs which do not have a link in different reference networks, using the Mann-Whitney U (MWU) test (Fig. 4a, right panel).

For all shortest path analyses, we observed that shortest path distances are significantly shorter for node pairs that are connected by a link in the reference networks (see Fig. 4c for a selection of the results). In particular, the results show (1) that distances between diseases that are connected by edges in diseasomes constructed based on

comorbidities, shared drugs, shared symptoms, or shared genetic variants are significantly shorter than distances between diseases without such edges (Supplementary Fig. 6a–d); (2) that distances of disease-drug pairs with shared indication edges are significantly shorter than distances of disease-drug pairs without such edges (Supplementary Fig. 6e); and (3) that distances between drug pairs with shared indication are significantly shorter than distances for drug pairs without shared indications (Supplementary Fig. 6f). In sum, our global analyses hence provide solid evidence for the global validity of the BEV network medicine paradigm and hence further corroborate the findings of previous studies[19–24].

## Results of local-scale analyses

To test the local-scale hypothesis, we computed *P*-values using the one-sided MWU test based on local GEDs to evaluate whether the local distances for the original networks are significantly smaller than the local distances for the permuted counterparts (Fig. 5a, left panel). Local GEDs of nodes obtained for the permuted and a selection of original networks and the corresponding MWU *P*-values are shown in

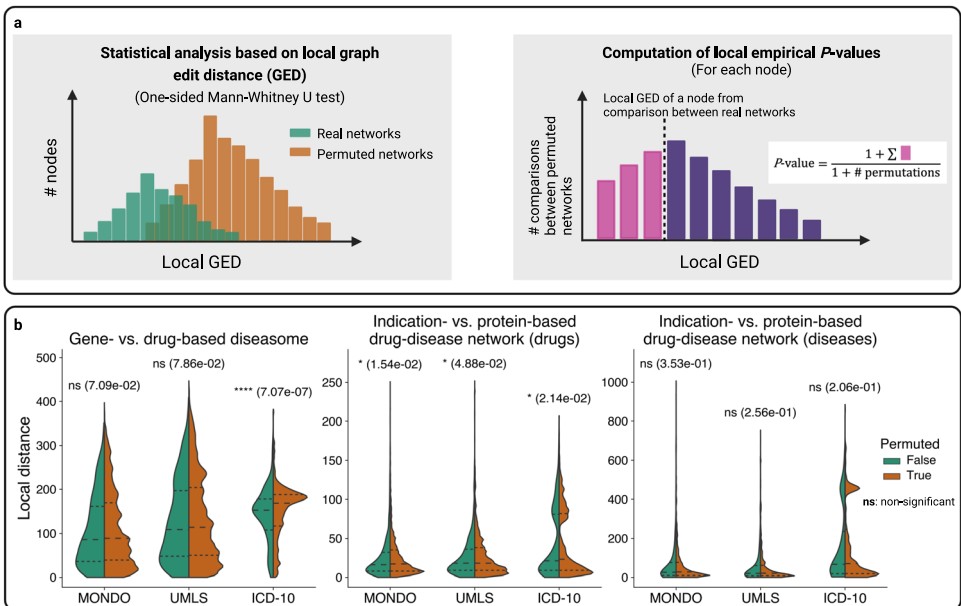

**Fig. 5 | Local-scale analyses: methods and local GEDs. a** Illustration of local-scale analysis methods. Left panel: Statistical analyses based on local GED via MWU test. Right panel: Computation of empirical *P*-values (one-sided, unadjusted) of each node based on local GEDs. **b** Local GEDs (of all nodes) between a selection of original networks vs. their permuted counterpart networks and corresponding MWU *P*-values. Left: Similarities between gene- and drug-based diseasome. Middle: Similarities between indication- and protein-based drug-disease network (for drugs). Right: Similarities between indication- and protein-based drug-disease network (for diseases). Results shown in (**b**) are based on uniform edge edit cost.

Fig. 5b (for the full results of the local-scale analyses, see Supplementary Fig. 7). The overview of the results of the local GED analyses in different vocabularies shows that the comparisons performed in ICD-10 vocabulary (at three-character level) led to more significant similarities than the ones performed in MONDO or UMLS CUI vocabulary (Fig. 6a and Supplementary Fig. 4a). As an example, the *P*-value computed from the local GEDs of drug-based vs. gene-based diseasomes in ICD-10 vocabulary is significant at 0.0001 level ($P \approx 7.1 \times 10^{-7}$), while it is not significant in the MONDO and UMLS CUI vocabularies ($P \approx 0.071$ for MONDO, $P \approx 0.079$ for UMLS CUI).

The results of the MWU test for local GED analyses point out that we have more significant similarities in ICD-10 (8 out of 10 significant at 0.05 level) than in MONDO vocabulary (2 out of 6 significant at 0.05 level) or UMLS CUI vocabulary (1 out of 6 significant at 0.05 level). The results also suggest that variant-based diseasomes have higher similarities with other diseasomes (7 out of 10 comparisons significant at 0.05 level) than gene-based diseasomes (5 out of 10 comparisons significant at 0.05 level), considering all three vocabularies. By inspecting the *P*-values of drug nodes (3 out of 3 comparisons significant at 0.05 level) against disease nodes (0 out of 3 comparisons significant at 0.05 level) obtained from local-similarity analyses of indication- versus protein-based drug-disease network as well as *P*-values obtained from target- and indication-based drugome (significant at 0.001 level), we discovered that, in general, drug neighborhoods are better preserved across the compared networks than disease neighborhoods (Fig. 6a, bottom right panel).

Furthermore, we computed local empirical *P*-values individually for nodes based on local GEDs (Fig. 5a, right panel). The local empirical *P*-values for all network comparisons are shown in Supplementary Fig. 8. The fractions of significant local empirical *P*-values at 0.05 level are shown in Fig. 6b and Supplementary Figs. 4b and 12b. Our results show that, for a substantial fraction of disease nodes, local neighborhoods are preserved not only not significantly better but worse than expected by chance across the different diseasomes (compare sigmoidal shape of curves in Supplementary Fig. 8). The local-scale hypothesis hence seems to hold for some diseases, but does not hold at all for others.

In follow-up analyses, we tried to identify patterns explaining these results, e.g., by assessing whether there are certain chapters of the ICD-10 disease vocabulary which are enriched with diseases with very small or very large empirical *P*-values. However, no clear patterns could be discovered, indicating that it is very hard to predict for which concrete diseases BEV network medicine approaches can be expected to yield robust and reliable results. Our local analyses hence only provide weak evidence for the local-scale hypothesis, indicating the BEV network medicine tends to produce locally blurred results.

### Web tool for interactive exploration of results

In order to make our results explorable and actionable, we developed the GraphSimViz (graph similarity visualizer) web interface, which is freely available at https://graphsimviz.net. GraphSimViz allows biomedical researchers to query and visualize our findings for user-selected drugs, diseases, network types, and disease vocabularies. Using GraphSimViz, biomedical researchers can assess if a specific type of disease association data is likely to contain reliable information about pathomechanisms underlying their diseases of interest. Below, we illustrate how GraphSimViz can be employed for interactive exploration of our results, using neurodegenerative diseases as a case example. To enable quantification of the effect of biases introduced by mechanistically ungrounded disease definitions in data sources not covered by our study, we provide the GraphSimQT (graph similarity quantification tool) Python package, which is freely available on GitHub (https://github.com/repotrial/graphsimqt).

### Discussion

Our results strongly support the global-scale hypothesis and, in line with previous studies[19–24], provide solid evidence for the overall validity of the BEV network medicine paradigm. However, they also indicate that results generated via BEV network medicine approaches become less reliable when zooming-in on individual diseases. Our results hence confirm that it is problematic to exclusively rely on data annotated with phenotype-based definitions if the objective is to uncover molecular pathomechanisms. As long as phenotype-based disease definitions have not been replaced by endotypes,

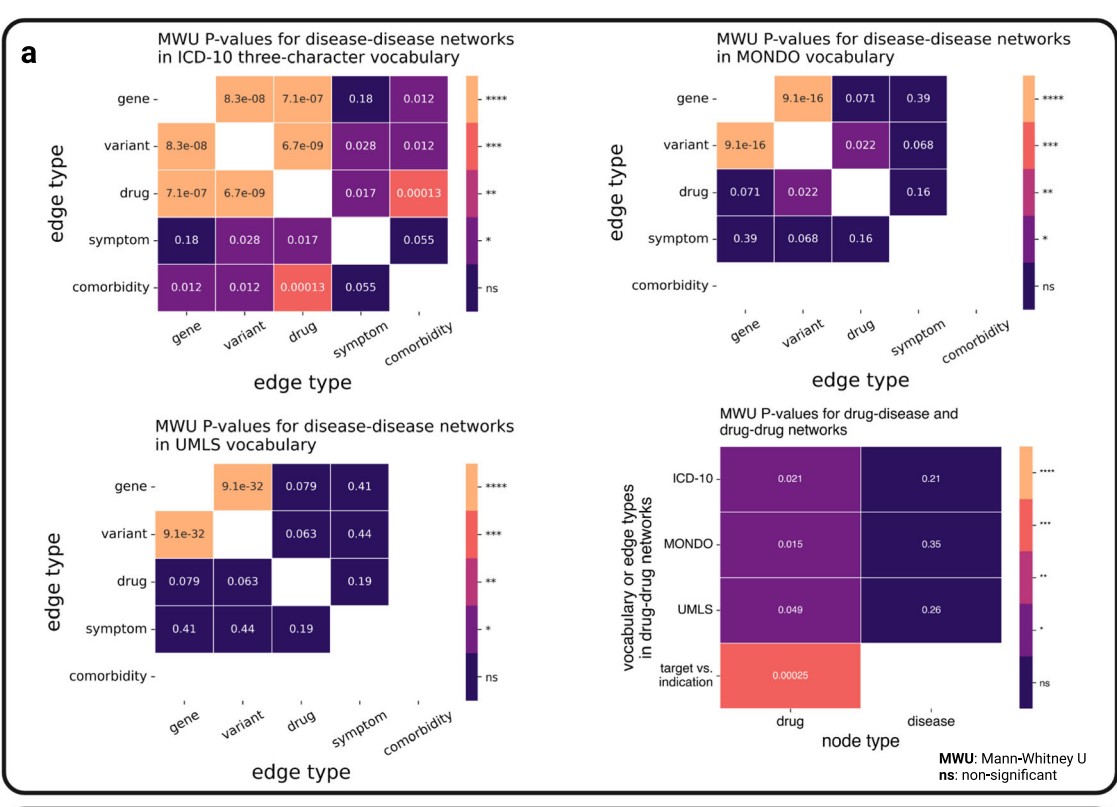

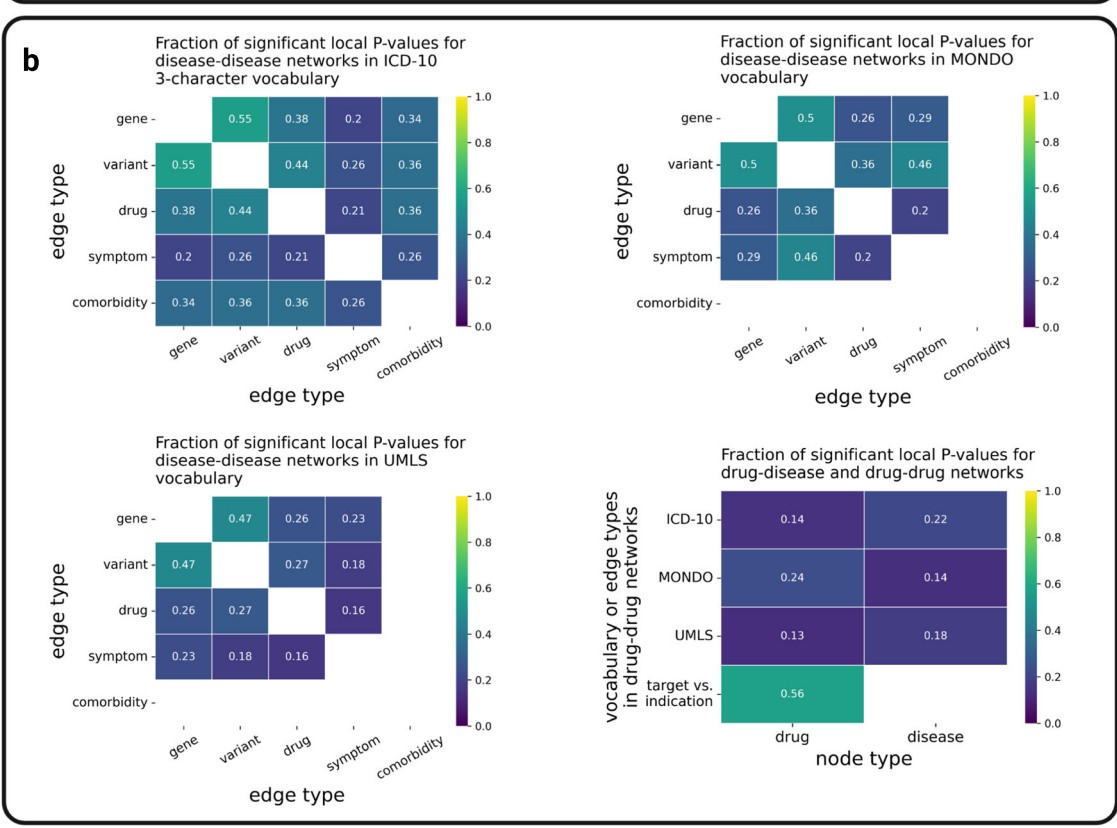

**Fig. 6 | Local-scale analyses: MWU *P*-values and local empirical *P*-values.**
**a** Overview of MWU *P*-values (one-sided, unadjusted) computed from local GEDs with levels of significance. **b** Fraction of significant local empirical *P*-values (one-sided, unadjusted) at 0.05 level computed from local GEDs on a pair of networks for the original vs. permuted network. All results are based on uniform edge edit cost.

large-scale disease association databases should therefore be used with care in network medicine and should be combined with additional layers of disease-specific omics data. In the following, we further speculate on issues that might play a role in the local

blurriness of BEV network medicine and sketch a roadmap to overcome this problem.

While there are vast amounts of datasets online that contain useful information about diseases such as genetic associations,

comorbidities, and symptoms, each of these datasets may use different disease vocabularies to describe their associations. The vocabularies have different degrees of granularity and are generated in different ways and for different purposes. However, for downstream (BEV) network medicine analyses, in order to jointly leverage the disease association from various data sources that use disease terms from different vocabularies as disease identifiers, we have to map data to a joint target vocabulary. This is a mammoth task that inevitably involves losing some data due to unmappable terms (see Fig. 7 for the levels of completeness of disease vocabulary mappings underlying this study).

The choice of the disease vocabulary has the potential to dramatically affect the results of downstream analyses (see discordant results of local-scale analyses carried out using ICD-10 three-character codes, on the one hand, and UMLS CUIs or MONDO IDs, on the other hand, shown in Fig. 6a and Supplementary Fig. 4a). At the same time, for most analysis tasks, the choice of the disease vocabulary is dictated by the format of the data and, thus, often impossible to change without losing information at the time of analysis. The vocabularies used to annotate disease-associated data must hence be viewed as confounders which are very difficult if not impossible to control for.

Currently used disease vocabularies are not only used discordantly, but also mechanistically inadequate: Since causal molecular disease mechanisms are often unknown, disease names often do not denote such mechanisms but rather reflect the person who coined the disease term (e.g., "Alzheimer's disease"), areas in the body that are affected (e.g., "kidney stones") or symptoms of the disease (e.g., "irritable bowel syndrome"). ICD-10 codes are considered inadequate due to their overly inclusive designations, ranging from symptoms (e.g., cough) over syndromes (e.g., cachexia) to true endotypes with definable molecular determinants (e.g., Mendelian disorders). This leads to data that is blurred, as diseases with distinct pathomechanisms are being aggregated together, e.g., due to symptom or organ commonality. This blurriness not only has severe clinical consequences (patients with mechanistically distinct diseases receive the same untargeted treatment), but also makes it very challenging to mine disease-associated data for pathomechanisms via BEV network medicine approaches[44]. Since such analyses often require case-versus-control or subtype annotations as input, it is very difficult to obtain meaningful results if the employed disease definitions are too unspecific.

The results presented in this study, where drugome comparisons have led to more significant results on a local level than diseasome comparisons, are evidence that network-based analyses yield more targeted and reliable results when the underlying annotations are well-defined (such as in drug vocabularies). Comparing the results of the GED-based analyses for full diseasomes (global analyses) with those obtained for analyses based on local GEDs in diseasomes with ICD-10 three-character codes, UMLS CUIs, and MONDO terms as nodes, respectively, further highlights the detrimental effect of local blurriness in currently used disease definitions: The higher the resolution of the analysis, the less significant the obtained $P$-values (see Fig. 8). When using MONDO or UMLS CUI terms (fine granularity) as nodes in the diseasomes, only the comparisons between gene- and variant-based diseasomes consistently (with respect to uniform, weight-based, and rank-based edit costs) led to smaller local distances in the original networks than in their randomized counterparts. No other network comparisons in the MONDO or UMLS vocabularies yielded significant $P$-values for all three types of edit costs. When using ICD-10 three-character codes (which denote disease clusters rather than individual diseases), around 50% of all computed MWU $P$-values are significant at 0.001 level. When comparing the entire diseasomes via global GEDs, all empirical $P$-values are significant.

The fact that we could not identify any clear patterns among diseases with small or large empirical $P$-values computed based on local GEDs may be a consequence of some of the current phenotype-based disease entities already corresponding to true endotypes. We speculate that, for diseases where our current definitions already have a one-to-one mapping to true endotypes, the local-scale hypothesis holds.

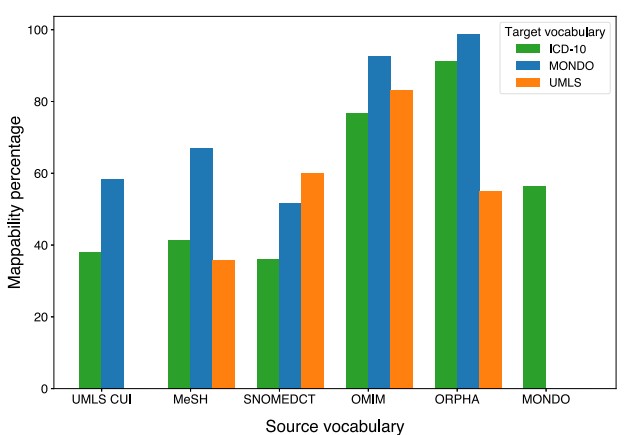

**Fig. 7 | Levels of completeness of disease vocabulary mappings underlying this article.** For each source-target vocabulary pair, mappability is computed as the percentage of terms in the source vocabulary used in this study that could be mapped to a term in the target vocabulary.

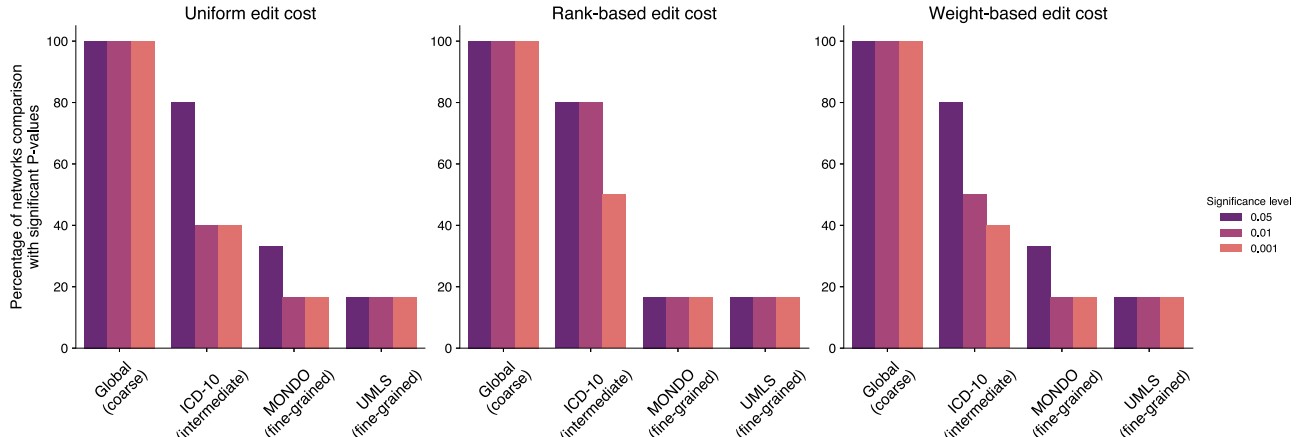

**Fig. 8 | Effect of disease term granularity on results of GED-based analyses.** For the individual $P$-values summarized in this figure, see Fig. 6a, as well as Supplementary Figs. 1, 4a, 5, 9, and 12a.

Even though we expected to obtain similar results for variant-based and gene-based diseasomes, the local-similarity analyses show that variant-based diseasomes have higher similarities with other diseasomes compared to gene-based diseasomes. This indicates that the disease-gene associations underlying the gene-based diseasomes contain less targeted information than the disease-variant associations underlying the variant-based diseasomes. Hence, using disease-variant data might yield more reliable results in the context of BEV network medicine applications.

To seek a possible explanation for this difference, we had a closer look at the associations underlying these two types of diseasomes. In our study, as well as in many other network medicine studies[1,2,22,45,46], disease-gene associations were taken from OMIM and DisGeNET curated databases. The latter collates disease-gene associations from different databases: UniProt[47], CTD[48], Orphanet[49], ClinGen[50], Genomics England[51], CGI[52], and PsyGeNET[53]. These constituent databases comprise multiple types of disease-gene associations such as causal mutations (mutations known to cause the disease), modifying mutations (mutations known to modify the clinical presentation of the disease), or merely statistical associations without evidence of causality. Disease-variant associations used in our study were extracted from DisGeNET, which itself integrates various databases: GWASdb[54], ClinVar[55], GWAS Catalog[56], UniProt, and BeFree[57]. Like for disease-gene associations, there are different types of disease-variant associations, ranging from known causal variants to variants with merely statistical evidence. However, the heterogeneity of the association types is higher for disease-gene associations than for disease-variant associations. Moreover, the genetic variation data from the constituent disease-variant databases of DisGeNET is mainly taken from genome-wide association studies (GWAS), which identify associations between common genetic variants and phenotypic traits via hypothesis-free, genome-wide scans. In contrast, in the disease-gene databases used by DisGeNET, parts of the data are curated from studies where evidence for disease-gene associations stems from a very limited number of patients or where hypothesis-driven approaches were used (i.e. the analyzed genetic variants were limited to those contained in candidate genes selected a priori).

Another reason for the difference in results between gene-based and variant-based diseasomes may consist in the loss of detail resulting from mapping variants to genes. Distinct mutations in one gene may cause different phenotypes, but this information cannot be captured at the level of disease-gene associations and is better conserved at disease-variant level. A very good example is the LMNA gene, where different mutations can cause 13 different diseases such as Hutchinson-Gilford progeria syndrome and the Dunnigan-type familial partial lipodystrophy[58]. Finally, the difference in results between gene- and variant-based diseasomes may also partly be due to loss of information introduced when aggregating $P$-values for disease-variant associations at gene level[59].

A limitation of our study is that our results do not rule out the possibility that confounders other than mechanistically inadequate disease definitions lead to the observed local blurriness of BEV network medicine. For instance, off-target effects might introduce biases in our analyses using drug association data, while the known biases in gene association data discussed above might explain the results obtained for analyses involving gene association data. However, we would like to stress that the obtained results are remarkably stable across all employed data modalities (see distributions of the obtained local empirical $P$-values in Supplementary Figs. 3, 8, and 11). Since phenotype-based disease definitions are the only confounders that affect all data types, this is strong (but of course not conclusive) evidence that the observed local blurriness can indeed mainly be attributed to them.

We started our investigation with the question of whether biases introduced by phenotype- and organ-based disease mechanisms even

out when mining large-scale disease association data for disease mechanisms – an assumption implicitly made by BEV medical research approaches. Our results indicate that this question has to be answered negatively, which has several consequences for the network medicine field and beyond.

Firstly, our findings imply that uncritical use of databases such as DisGeNET or OMIM which rely on phenotype-based disease definitions is problematic. Instead, we emphasize that close-up approaches remain the gold standard in network medicine, where data scientists collaborate with researchers from the biomedical sciences and jointly analyze molecular as well as deep phenotype data for the same patients. In such a collaborative setup, a positive feedback loop can emerge, where initial hypotheses about disease subtypes and their underlying pathomechanisms are formulated based on the analysis of molecular data, further refined using deep phenotyping (e.g., histological images, blood-derived biomarkers, etc.) and expert knowledge of the clinicians, and finally validated in preclinical studies (e.g., gain- or loss-of-function studies). As mentioned above, such approaches have already led to various important insights into specific disease mechanisms.

Secondly, unsupervised network medicine methods are needed, which not only return candidate pathomechanisms but at the same time de novo stratify patients into mechanistically distinct subgroups and hence do not rely on potentially misleading priorly available phenotypically defined subtype annotations. While few such approaches exist[60–62], most existing pathomechanism mining methods still rely on phenotypic case-versus-control annotations[63,64] or lists of genes associated with a (potentially ill-defined) disease term[65–67].

Finally, we would like to point out that the current lack of mechanistic disease definitions not only hampers progress in (BEV) network medicine, but also has a detrimental effect on virtually all other data-centric approaches to, e.g., treatment design or diagnosis which rely on disease association data that utilize phenotype-based disease definitions. For instance, an artificial intelligence model for diagnosis assistance trained on genetic disease signatures will systematically produce unreliable results if the disease annotations used for training do not correspond to true endotypes. While we here quantified the effect of this problem in the context of BEV medicine, overcoming it would hence be beneficial for a large fraction of the biomedical research community.

## Methods

### Compliance with ethical regulations

Our research complies with all relevant ethical regulations. The only non-public data used for this study is the comorbidity data we obtained from the Estonian Biobank. The Estonian Biobank is a population-based biobank managed by the Institute of Genomics at the University of Tartu. All participants have signed a broad consent upon joining the biobank, allowing their sample and data to be used for further research. ICD-10 diagnoses are obtained from epicrises, prescriptions and bills to the Health Insurance Fund. The work in this article was covered by the ethics approval "234T-12 Omics for Health" (March 19, 2014) by the Estonian Committee of Bioethics and Human Research. Data was released by the Estonian Biobank (release M11, July 24, 2019).

### Data integration

As shown in Table 1, the data sources used to create the different networks use a range of competing disease vocabularies to refer to diseases. We hence had to map these vocabularies to a common vocabulary to be able to investigate network (dis-)similarities. The similarity analyses were performed in MONDO (Monarch Disease Ontology), UMLS CUI, and ICD-10 vocabularies. Disease ID mapping to MONDO and ICD-10 was carried out via the two-step approach implemented in the NeDRex platform[68]: First, MONDO contains mappings between its own disease vocabulary and various other

**Table 1 | Data sources used for network construction**

| Data source | Used disease vocabularies | Data type | Networks constructed from data source |
|---|---|---|---|
| HPO[86] | OMIM, Orphanet (ORPHA) | Disease-symptom | Symptom-based diseasome |
| DisGeNET | Concept Unique Identifiers of Unified Medical Language System (UMLS CUI) | Disease-gene, disease-variant | Gene-based diseasome, variant-based diseasome, disease-gene-gene-disease network, drug-protein-protein-drug network, drug-protein-protein-disease network |
| OMIM | OMIM | Disease-gene | Gene-based diseasome, disease-gene-gene-disease network, drug-protein-protein-disease network |
| DrugCentral[37] | SNOMED Clinical Terms[87] (SNOMEDCT) | Drug-target, drug-indication | Target-based drugome, indication-based drugome and drug-disease network, drug-protein-protein-drug network, drug-protein-protein-disease network |
| DrugBank[88] | – | Drug-target | Target-based drugome, drug-protein-protein-drug network, drug-protein-protein-disease network |
| CTD[48] | MeSH | Drug-indication | Drug-disease network, indication-based drugome |
| IID[89] | – | Protein-protein interaction | Disease-gene-gene-disease network, drug-protein-protein-drug network, drug-protein-protein-disease network |
| UniProt | – | Gene-protein | Drug-protein-protein-disease network |
| Estonian Biobank[90] | ICD-10 (mixed three- and four-character codes) | Comorbidity data | Comorbidity-based diseasome |

vocabularies, including OMIM, MeSH[69], and ICD-10. Then, mappings between several vocabularies and ICD-10 could be achieved by mapping disease terms to MONDO, followed by mapping MONDO to ICD-10. Mapping to UMLS CUI was carried out using the mappings provided in the UMLS Metathesaurus 2022AA full release. For all pairwise analyses, the two compared networks were aligned before computing GEDs, i.e., only the nodes contained in both of them were taken into account.

The comorbidity data was obtained from the Estonian Biobank, which uses originally ICD-10 codes. In order to carry out analyses involving comorbidity data in MONDO or UMLS CUI vocabulary, the comorbidity data needed to be mapped from a coarser-grained (ICD-10) to a finer-grained disease vocabulary (MONDO and UMLS CUI). Although this is possible from a technical point of view, it would have introduced a lot of noise in the obtained comorbidity networks. To avoid overshadowing all other effects by the introduced noise, we decided to carry out analyses involving comorbidity data only in ICD-10 vocabulary. Consequently, all analyses involving comorbidity data were carried out only in ICD-10 vocabulary. On the other hand, the comparison between the target- and the indication-based drugomes was carried out only in MONDO vocabulary. In these networks, nodes are drugs and not diseases and using different disease vocabularies leaves the nodes of the networks unchanged. In the indication-based drugomes, the choice of the disease vocabulary can change the edges of the networks, but, in practice, we observed that the differences are small. Target-based drugomes are not affected at all by the choice of the disease ontology. Therefore, we only use MONDO for the comparison of drugomes.

Additionally, further data harmonization steps were carried out: Since HPO contains both general and specific terms, we pruned the data by removing very general symptom terms, using the existing hierarchy in HPO. More specifically, we decomposed the generated hierarchical phenotype network into its levels and removed the terms from the top three levels.

The diagnoses in around 140 K patients records available in the Estonian Biobank (April 2020 version used for this study) are encoded in ICD-10 vocabulary, and the records contain both three- and four-character ICD-10 codes. In order to generate uniform data, we therefore truncated all four-character codes to three-character level. Moreover, we removed diseases with incidence below five from the data, as well as the codes from the ICD-10 chapters XV ("Pregnancy, childbirth and the puerperium"), XVI ("Certain conditions originating in the perinatal period"), XVIII ("Symptoms, signs and abnormal clinical and laboratory findings, not elsewhere classified"), XIX ("Injury,

poisoning and certain other consequences of external causes"), XX ("External causes of morbidity and mortality"), XXI ("Factors influencing health status and contact with health services"), and XXII ("Codes for special purposes").

### Network construction

For network construction, some part of the data such as disease-gene, drug-indication, drug-target, gene-encoding-protein, and PPI data were obtained from the databases shown in Table 1, using the data access and mapping provided by the NeDRex platform[68]. Disease-variant and disease-symptom associations were directly obtained from DisGeNET and HPO, respectively.

Supplementary Table 1 shows the most important properties of all constructed networks. The comorbidity-based diseasome was constructed via $\phi$-correlation. Let $I_i$ denote the incidence of disease $i$ and $C_{ij}$ be the number of patients who were simultaneously diagnosed with diseases $i$ and $j$. The comorbidity between the two diseases can be measured by

$$\phi_{ij} = \frac{C_{ij}N - I_iI_j}{\sqrt{I_iI_j(N - I_i)(N - I_j)}}, \tag{1}$$

where $N$ is the total number of patient records ($N = 139,065$ for the Estonian Biobank data). When two diseases co-occur more frequently than expected by chance, we have $\phi_{ij} > 0$. We used one-tailed Fisher's exact test followed by Benjamini-Hochberg correction for multiple testing to determine the significance of comorbidity associations and connected two diseases by an edge if adjusted $P \le 0.05$. Edge weights were defined using the $\phi$-correlation, i.e., we set $w_{ij} = \phi_{ij}$ for all diseases $i$ and $j$ with significant comorbidity association.

The indication- and target-based drugomes as well as the gene-, variant-, symptom-, and indication-based diseasomes were constructed based on the Jaccard index of the respective annotations. $A_i$ denotes the set of annotations for a disease or drug $i$ used as node in the network under construction (e.g., when constructing the gene-based diseasome, $A_i$ is the set of all genes associated with disease $i$). We connected diseases $i$ and $j$ by an edge if $|A_i \cap A_j| \ge 1$ and defined the edge weights as $w_{ij} = |A_i \cap A_j|/|A_i \cup A_j|$. Disease nodes with $|A_i| = 0$ were removed from the networks, i.e., empty annotation sets were treated as missing data.

The bipartite indication-based drug-disease network was directly constructed from the data source, i.e., we connected a disease $i$ with a drug $j$ if $i$ is an indication for $j$. For the bipartite target-based drug-

disease network, we connected a disease $i$ with a drug $j$ if $j$ targets a protein encoded by a gene associated to $i$. In both drug-disease networks, edges are unweighted. Finally, we constructed drug-protein-protein-disease networks where drugs are connected to their targets, experimentally validated PPIs from IID are used to connect proteins, and diseases are connected to proteins encoded by disease-associated genes.

## Graph edit distance

GED is a widely used and generically applicable distance measure for attributed graphs[38,39,70]. It is defined as the minimum cost of transforming a source graph $G_1 = (V_1, E_1)$ into a target graph $G_2 = (V_2, E_2)$ via elementary edit operations, i.e., by deleting, inserting, and substituting nodes and edges. Equivalently, GED can be defined as the minimum edit cost induced by a node map $\pi$ from $G_1$ to $G_2$, where nodes maps $\pi \subseteq (V_1 \cup \{\epsilon_1\}) \times (V_2 \cup \{\epsilon_2\})$ are relations that cover all nodes $u \in V_1$ and $v \in V_2$ exactly once ($\epsilon_1$ and $\epsilon_2$ are dummy nodes that may be covered multiple times or left uncovered)[71].

We used a customized version of GED to compare the different diseasomes, drugomes, and drug-disease networks constructed as detailed in the previous section as well as their randomized counterparts. Since the networks were aligned before all pairwise comparisons, we had $V_1 = V_2 = V$ (node sets are identical) whenever comparing two networks. Consequently, we fixed $\pi$ as the identity and computed GED as the sum of edge edit costs induced by the identity (the edge edit cost functions sub, del, and ins are explained below):

$$\text{GED}(G_1, G_2) = \sum_{uv \in E_1 \cap E_2} \text{sub}(uv) + \sum_{uv \in E_1 \setminus E_2} \text{del}(uv) + \sum_{uv \in E_2 \setminus E_1} \text{ins}(uv) \quad (2)$$

$\text{GED}(G_1, G_2)$ quantifies the global distance between the graphs $G_1$ and $G_2$. Since the node sets of $G_1$ and $G_2$ are identical in our analyses, it can be decomposed as

$$\text{GED}(G_1, G_2) = \sum_{u \in V} \text{GED}(G_1, G_2, u)/2, \quad (3)$$

where $\text{GED}(G_1, G_2, u)$ is the local distance between the neighborhood $N_1(u)$ of node $u$ in $G_1$ and its neighborhood $N_2(u)$ in $G_2$. The local distances are defined as follows:

$$\text{GED}(G_1, G_2, u) = \sum_{v \in N_1(u) \cap N_2(u)} \text{sub}(uv) + \sum_{v \in N_1(u) N_2(u)} \text{del}(uv) + \sum_{v \in N_2(u) N_1(u)} \text{ins}(uv) \quad (4)$$

Based on the local distances, we also computed cluster-level distances for a cluster of nodes $C \subseteq V$ as follows:

$$\text{GED}(G_1, G_2, C) = \sum_{u \in C} \text{GED}(G_1, G_2, u)/2 \quad (5)$$

We used three types of edge edit cost functions, namely, uniform costs and costs based on normalized edge ranks or normalized edge weights. The uniform costs are defined by simply setting $\text{sub}(uv) = 0$ and $\text{del}(uv) = \text{ins}(uv) = 1$ for all edges $uv$. GED with uniform costs quantifies topological (dis-)similarity between two graphs but does not consider edge weights. Since edges are weighted in all compared diseasomes, we additionally defined edge edit costs based on normalized weights and normalized ranks. For the normalized weights, we scaled all edge weights to the interval [0,1] via division by the maximum. For the normalized ranks, we sorted the diseasomes' edges in increasing order with respect to their weights and then again normalized the obtained ranks to [0,1] via division by the maximum rank. Let $x_1(uv)$ be the normalized weight/rank of edge $uv$ in diseasome $G_1$ and $x_2(uv)$ be its normalized weight/rank in $G_2$. Then we defined the weight-/rank-based edit costs as $\text{sub}(uv) = |x_1(uv) - x_2(uv)|$, $\text{del}(uv) = x_1(uv)$, and $\text{ins}(uv) = x_2(uv)$. That is, substitutions are expensive if the involved edge's normalized weight/rank differs a lot in the two graphs and deletions and insertions are more expensive for high-weighed/low-weighed/low-ranked edges. Since uniform, weight-based and rank-based edit costs led to similar results, we only present the results for uniform costs in the main article. Results for weight- and rank-based edit costs are shown in the supplement.

## Statistical analyses based on graph edit distances

Using GED, we tested the local- and the global-scale hypotheses as follows: For each pair $G_1$, $G_2$ of compared networks, we generated 1,000 randomized counterparts $G_1^1, \ldots, G_1^{1000}$ and $G_2^1, \ldots, G_2^{1000}$. For this, we used a random network generator which repeatedly swaps edges and non-edges to obtain randomized counterparts which exactly preserve the node degrees of the original networks[72,73]. For each node $u$, we then computed $\text{GED}(G_1, G_2, u)$ as well as $\text{GED}(G_1^i, G_2^i, u)$ for each $i = 1, \ldots, 1000$ and also computed the global distances $\text{GED}(G_1, G_2)$ and $\text{GED}(G_1^i, G_2^i)$.

To test the global-scale hypothesis, we computed one-sided empirical $P$-values as

$$P = \left(1 + \sum_{i=1}^{1000} \left[\text{GED}(G_1, G_2) \ge \text{GED}\left(G_1^i, G_2^i\right)\right]\right)/(1 + 1000), \quad (6)$$

where [true] = 1 and [false] = 0. To test the local-scale hypothesis, we used the one-sided MWU test to assess whether the local distances $\{\text{GED}(G_1, G_2, u) | u \in V\}$ for the original networks are significantly smaller than the local distances $\{\text{GED}(G_1^i, G_2^i, u) | u \in V, i = 1, \ldots, 1000\}$ for the randomized counterparts. Moreover, we computed node-specific local empirical $P$-values as

$$P(u) = \left(1 + \sum_{i=1}^{1000} \left[\text{GED}(G_1, G_2, u) \ge \text{GED}\left(G_1^i, G_2^i, u\right)\right]\right)/(1 + 1000) \quad (7)$$

and cluster-level empirical $P$-values as

$$P(C) = \left(1 + \sum_{i=1}^{1000} \left[\text{GED}(G_1, G_2, C) \ge \text{GED}\left(G_1^i, G_2^i, C\right)\right]\right)/(1 + 1000) \quad (8)$$

where $C \subseteq V$ is a cluster of nodes.

Note that we consciously refrained from adjusting $P$-values for multiple testing. The reason for this choice is that the relevance of our results stems from the non-significance of a large fraction of the obtained $P$-values. If we had corrected for multiple testing, we would have inflated this fraction.

## Rationale for using the graph edit distance as a measure of network dissimilarity

In addition to our version of GED, there are various other network dissimilarity measures—most notably, embedding-based[74,75], kernel-based[76], and message-passing-based[77,78] approaches. We decided to use GED because, to the best of our knowledge, it is the only distance measure satisfying the following requirements necessary for our analyses:

1. To allow testing both the global- and the local-scale hypothesis, we need a graph distance measure $d(G_1, G_2)$ which is decomposable into local node distances $d(G_1, G_2, u)$.
2. The local node distances $d(G_1, G_2, u)$ should depend on $u$'s local neighborhoods in $G_1$ and $G_2$ but not on the overall network topologies (otherwise, we would not be testing the local-scale hypothesis when comparing local node distances).
3. Since a node alignment between the compared networks is given (disease and drug terms are aligned between the networks), both the global network distance $d(G_1, G_2)$ and the local distances $d(G_1, G_2, u)$ should be node-identity-aware rather than permutation-invariant.
4. The distances need to be computable in linear time w.r.t. the size of the networks in order to enable our large-scale permutation tests.

While most of the kernel-based methods already fall short of requirement 1, popular node-embedding-based approaches (e.g.,

node2vec[74] with subsequent distance computation in embedding space) typically do not satisfy requirements 2 through 4. Exceptions we are aware of are DeltaCon[79] (which satisfies requirements 1, 3, and 4 but not requirement 2) and the graphlet degree signature[80] (which satisfies requirements 1 and 2 but not requirements 3 and 4). Highly successful techniques in graph learning follow a message passing concept[77,78]. When restricted to a single hop (as needed to satisfy requirement 2), these methods define node $u$'s embedding in the graph $G_1$ as $x_1(u) = g(\{l(v)|v \in N_1(u)\})$, where $l(v)$ is the label of node $v$ (its disease or drug term) and $g$ is a permutation-invariant function[77] mapping sets to vectors (e.g., indicator function). Here, using unique node labels renders the method node-identity-aware and allows to drop $l(\cdot)$ as a parameter of $g$. Such approaches fulfill all four requirements, but are essentially equivalent to GED with uniform edge costs: By comparing $u$'s embeddings $x_1(u)$ and $x_2(u)$, we compare the node labels of its neighboring nodes in $G_1$ and $G_2$, which is exactly what we do with uniform GED.

## Statistical analyses based on shortest path distances

We carried out analyses based on shortest path distances between (1) all disease-disease pairs in a disease-gene-gene-disease network, (2) all drug-drug pairs in a drug-protein-protein-drug network, and (3) all disease-drug pairs in a disease-protein-protein-drug network. For each network, we split the multi-set of obtained distances into multi-sets $X_0$ and $X_1$, where $X_1$ contains the shortest path distances for all nodes pairs contained as edge in a reference network and $X_0$ contains all other shortest path distances. As reference networks, we used (1) drug-, symptom-, comorbidity-, and variant-based diseasomes, (2) a bipartite drug-indication network, and (3) an indication-based drug-drug network. We then used the one-sided MWU test to assess whether the shortest path distances contained in $X_1$ are significantly smaller than those contained in $X_0$.

## BEV network medicine is committed to the local- and the global-scale hypotheses

Recall that we have introduced BEV network medicine as the subfield of network medicine which aims at uncovering disease mechanisms by mining large-scale disease-association data. Let $D_1$ be data used towards this end by BEV network medicine approaches and let $d_1$ and $d_2$ be two diseases sharing an (unknown) molecular mechanisms $M$ such that $D_1$ contains entries $D_1(d_1)$ and $D_1(d_2)$. If $D_1$ contains any useful information about disease mechanisms as assumed by BEV network medicine, $M$ should lead to significant similarities between $D_1(d_1)$ and $D_1(d_2)$. The same holds for any other data $D_2$ used as input by BEV network medicine. BEV network medicine is hence implicitly committed to the claim that the edge distributions of diseasomes $G_1$ and $G_2$ constructed based on similarities in $D_1$ and $D_2$ exhibit a higher correlation than expected by chance. This, in turn, implies both the global- and the local-scale hypothesis.

## Implementation

We have implemented all network analysis approaches underlying this article in a Python package called GraphSimQT. GraphSimQT uses graph-tool[81] for network handling and Scipy[82] for carrying out statistical tests and comes with all networks and scripts to reproduce the results reported in this paper. Moreover, GraphSimQT can be used to compare user-provided networks, using the techniques presented in this paper. Significance of comorbidity associations was evaluated using the Scipy implementation of Fisher's exact test and the statsmodels[83] implementation of Benjamini-Hochberg multiple testing correction. The GraphSimViz web tool (https://graphsimviz.net) was implemented using Vue.js as a frontend framework, the Drugst.One (https://drugst.one) plugin as network explorer and a Django backend with a PostgreSQL database.

## Reporting summary

Further information on research design is available in the Nature Portfolio Reporting Summary linked to this article.

## Data availability

All networks underlying the findings of this study are available at https://doi.org/10.5281/zenodo.7498864. The following public databases were used to generate the networks: IID (http://iid.ophid.utoronto.ca/), DrugBank (https://go.drugbank.com/), DrugCentral (https://drugcentral.org/), CTD (http://ctdbase.org/), DisGeNET (https://www.disgenet.org/), OMIM (https://omim.org/), UniProt (https://www.uniprot.org/), MONDO (https://MONDO.monarchinitiative.org/), NeDRex (https://nedrex.net/), and HPO (https://hpo.jax.org/app/). Version numbers of all used databases can be found in an AIMe report[84] for our study (https://aime.report/6bdnlg). The comorbidity-based diseasome was constructed based on data provided by the Estonian Biobank (https://genomics.ut.ee/en/content/estonian-biobank, available from the Estonian Biobank upon request). The construction of the networks is described in the Methods section of this paper. Our study is based on public databases (including DisGeNET, OMIM, DrugBank, HPO, and more) which do not contain sex-specific information. Therefore, no sex-specific analyses could be carried out. Source data are provided in this paper. They can also be downloaded from https://api.graphsimviz.net/download_results. Source data are provided with this paper.

## Code availability

The GraphSimQT tool is available at https://github.com/repotrial/graphsimqt, together with scripts to reproduce all results reported in this article. A stable version is available from Zenodo[85] (https://doi.org/10.5281/zenodo.7498864). The source code of the frontend and the backend of GraphSimViz is available at https://github.com/repotrial/GraphSimViz-frontend and https://github.com/repotrial/GraphSimViz-backend, respectively.

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

## Acknowledgements

S.S., E.A., J.S., A.W., and J.B. are grateful for financial support from REPO-TRIAL. REPO-TRIAL has received funding from the European Union's Horizon 2020 research and innovation programme under grant agreement No 777111. This publication reflects only the authors' view and the European Commission is not responsible for any use that may be made of the information it contains. This work was supported by the German Federal Ministry of Education and Research (BMBF) within the framework of the e:Med research and funding concept (grant 01ZX1908A) (S.S. and J.B.). J.B. was partially funded by his VILLUM Young Investigator Grant no. 13154. N.M.K. was supported by the Vienna Science and Technology Fund (WWTF) through project VRG19-009. The work of J.K. was supported by the Estonian Research Council grant PUT (PRG1291). T.H. was supported by the European Union through the European Regional Development Fund (Project No. 2014-2020.4.01.15-0012). The Estonian Biobank was funded by the European Union through the European Regional Development Fund Project No. 2014-2020.4.01.15-0012 GEN-TRANSMED. Data analysis by J.K. and T.H. was carried out in part in the High-Performance Computing Center of University of Tartu. The Estonian Biobank Research Team includes Mari Nelis, Lili Milani, Tõnu Esko, Andres Metspalu, and Reedik Mägi. Figures 1–6 and Supplementary Fig. 4 and 12 were created with BioRender.com.

## Author contributions

D.B.B. and S.S. conceived and designed this study and implemented the GraphSimQT Python package to compare the different networks. S.S. carried out the analyses. J.S., S.S., and K.A. integrated the data and constructed the networks. A.Ma. implemented the GraphSimViz web

tool. D.B.B., S.S., E.A., N.M.K., and A.Mo. drafted the manuscript. J.K., T.H., and the Estonian Biobank Research Team provided the comorbidity data. D.B.B., J.B., and A.W. supervised the project. All authors provided critical feedback and discussion and assisted in interpreting the results and writing the manuscript.

## Funding

## Competing interests
The authors declare no competing interests.
