## [Peer Review File · Nature Communications]

Reviewers' Comments:

Reviewer #1:

Remarks to the Author:

Sadegh et al. investigate the problem of data incompleteness and multi-dimensional bias in large gene-disease association datasets. Contributions in this manuscript are two-fold.

First, the manuscript puts forward an observation that prevailing global network medicine approaches make an implicit assumption that does not hold in the analysis of real-world datasets. The methods assume biases in gene-disease association datasets (due to phenotype-based disease definitions) can even out. Further, the approaches assume that, despite those biases, these datasets contain helpful information about the disease pathomechanisms to be uncovered.

Second, the manuscript quantifies to which extent data backs the assumption mentioned above. To this end, Sadegh et al. constructed diseasomes based on (1) disease-gene associations, (2) disease-variant associations, (3) comorbidity data, (4) symptom data, and (5) drug-indication data. They also constructed drug-disease and drug-drug networks based on drug indication and drug-target information. They then compared all pairs of diseasomes, drugomes, and drug-disease networks on a global and local scale using graph edit distance. Sadegh et al. also evaluated how competing disease ontologies of different granularity affect the results, analyzing MONDO IDs (finer granularity) and ICD-10 three-character codes (coarser granularity) as node IDs in the constructed networks, respectively.

Overall, the manuscript is well-written and easy to follow. However, I have several concerns that I'd like to see addressed before the paper is further considered for publication.

1) The importance of data incompleteness and data bias in gene (or variant) disease associations is not new. For example, Menche et al. have studied this issue comprehensively in <https://www.science.org/doi/10.1126/science.1257601>. The investigative bias in function and disease genomics datasets has been studied in the past using various perspectives, for example, Stoeger et al., "Large-scale investigation of the reasons why potentially important genes are ignored" (PLoS Biology 2018); Haynes et al. "Gene annotation bias impedes biomedical research" (Scientific Reports 2018); Esteban et al., "Information silos distort biomedical research" and "The speed of information propagation in the scientific network distorts biomedical research"; Kustatscher et al., "Understudied proteins: opportunities and challenges for functional proteomics" (Nature Methods 2022) and other papers, some of which are cited by the aforementioned articles.

2) Considering earlier research already pointing to the issue of data incompleteness and bias, the novelty of the observation put forward in this manuscript is relatively limited. That would not be an issue on its own if the manuscript would quantify the extent of the bias rigorously or develop a novel theory (see my concern #4). It is unclear to which extent the specific modeling decisions made by the authors influenced the results. For example, the study uses customized versions of the graph edit distance (GED) to compare networks. However, there are many other types of network metrics to compare networks. Why is the selected approach the most appropriate? Further, the analysis is done using MONDO and ICD-10 codes linked with the OMIM and DisGeNet namespace. How have the authors resolved disease vocabularies (e.g., disease subtypes, uncertain disease matching)? In the analysis of drug-related data, other confounders that could attenuate or emphasize data bias are not discussed. For example, the extent of data bias can depend on therapeutic area, molecular scaffolds, drug approval date, therapeutic modality, and other biological and biochemical properties, such as chemical classification and MoA, beyond the ATC classification considered in the paper. Are there specific subsets of diseases, drugs, and proteins for which bias is more substantial or weaker? Can the authors disentangle the contributions of various cofounders by defining an appropriate causal model and using partial correlation?

3) Findings of this study imply that uncritical use of DisGeNET or OMIM databases, which rely on phenotype-based disease definitions, is problematic. How do those findings generalize to other prominent sources of gene-disease association data available in the biomedical community?

4) My biggest concern with the study is that the key results are not actionable, in the sense that the study does not offer a statistically-grounded strategy to a) quantify the bias in a new study that could help researchers decide what BEV/close-up analyses are appropriate vs. not OR b) mitigate the bias identified. The study writes that "disease association databases should hence be used with care." However, it is unclear what this means and what concrete statistical/network science/information theoretic technique the study provides to help others use the databases "with care." Rigorous theoretical or statistical grounding or definitions of these concepts is lacking.

Reviewer #2:

Remarks to the Author:

The authors address an important question: to which degree are results obtained in the network medicine field reliable or rather the result of inherent biases in the way the underlying networks are generated. In particular, networks are generated with phenotype-based definitions of gene-disease relations, which the field actually wants to overcome. The authors conclude that network medicine approaches allow for an accurate global view of diseases and their network neighborhood but when zooming in the local context information is less accurate.

The observations are very important for the network medicine community (and beyond) as they make a convincing point for the need to handle diseaseome's generated from phenotype-based disease definitions with care. The methodology is sound and clearly supports the claims of the manuscript.

Specific comments:

1. I would aim to summarize and interpret the observations a bit more towards their clinical relevance or the way the disease association data is generated. Eg for which diseases is the local neighborhood pairwise more similar across different diseaseomes? Do they belong to specific groups of diseases? Could differences between diseases reflect different degree of biases, number of disease genes, degree of the disease node etc?

2. Overall enough detail is given to understand the methodology. However, It would be helpful to explain in a few words the way the local- and global-scale hypotheses are tested already a bit earlier in the manuscript (ie in the Introduction where the hypothesis are explained or in the beginning of the Results section) to avoid too much jumping back and forth between Methods and other sections for the reader.

3. The manuscript is well written but there are a few typos. Eg in line 560 there is a missing space before the P.

Reviewer #3:

Remarks to the Author:

The authors postulate that a lack of mechanistic disease definitions and related data impacts network medicine progress. Further, proposing that phenotype-based disease classification systems be replaced by endotypes that define distinct molecular mechanisms driving disease phenotypes, and that a bird's eye view network medicine approach would identify novel insights into disease mechanisms.

Certainly, the inclusion molecular mechanism-based endotypes would enhance the identification of connections between diseases.

The Introduction mentioned an endotype based ontology as an ideal outcome of this work, in order to achieve this objective, network medicine "approaches aim at uncovering pathomechanisms driving diseases".

-Background information on the scope of diseases that are defined by phenotype-based definitions and that also have known molecular mechanisms would have strengthened the premise of this

analysis.

- This statement, in the Introduction, is unsupported by literature.

(line 139) The biases current disease definitions introduce in large-scale disease association databases such as OMIM and DisGeNET do not even out and such databases should hence be used with care in all fields of data-centric biomedicine.

-The rationale for the analysis is based on unsupported assumptions: that phenotype-based definitions should and could be replaced by molecular-mechanism based definitions; which assumes that all phenotype-defined diseases have an known underlying molecular mechanism.

-The study states that they are comparing two ontologies, however, that is not the case. The article compares the mondo ontology with icd-10, a clinical vocabulary developed for morbidity and mortality code. Mondo is an ontology that was built from combining other disease ontologies along with clinical vocabularies, including ICD-10. ICD-10, is a vocabulary, does not include the semantic relationships between disease classes. The choice of these vocabularies may have biased the results, the as the two sources are not distinct.

-The analysis based on only two datasets (mondo and icd-10) limits the conclusions of the study. For some of the analysis only one ontology/vocabulary was utilized. See:

(line 183) Note that analyses involving comorbidity data were carried out only in ICD-10 and the comparison between target- and indication-based drugomes only in MONDO namespace.

The literature support for the statements, below, are not provided in the manuscript.

(lines-87- 93):

"Let D be disease association data of some data type T commonly used by BEV approaches (e.g., disease-gene associations). Further assume that D contains entries D (d1) and D (d2) for two diseases d1 and d2 that share an unknown molecular disease mechanism. Then this shared mechanism should lead to similarities between D (d1) and D (d2), given that D indeed contains useful information about disease mechanisms."

"For instance, we would expect that the diseases d1and d2 have similar profiles of disease-associated genes, that they exhibit high comorbidity, that they lead to similar symptoms, and that they can be treated by similar drugs.

-For most of the figures, the font size is too small for readability, and several figures are missing axis labels

(starting at line 307)

In some areas of the manuscript, such as the "Discordant disease ontologies", statements are not supported. Nor are the vocabularies referenced. The statement that these vocabularies are discordant is misleading.

Figure 6 (Levels of completeness of disease ontology mappings underlying this article) incorrectly labels these vocabularies as ontologies, does not recognize that they have been built for different purposes and thus it is reasonable that they may have non-overlapping disease terms, for example OMIM defines Mendelian genetic phenotypes, MESH includes terms from PubMed, and orphanet is a vocabulary of European rare disease. The UMLS_CUI represents the National Library of Medicine's Unified Medical Language System, which maps disease concepts between SNOMED, ICD, MESH and OMIM, among other vocabularies.

This figure has not included ICD-10 as one of the Source Ontologies.

The statement that 'Disease names are variable and non-standardized' (line 328) - is not an accurate statement, disease names are standardized and follow community guidelines. This statement misrepresents the history of disease nomenclature.

The presentation of the data figures makes it difficult to view and interpret the outcome of the

analysis.

Response letter for manuscript “Lacking mechanistic disease definitions and corresponding association data hamper progress in network medicine and beyond”

Summary of changes

We would like to thank the reviewers for their very constructive comments which helped us to substantially improve our study. The most important changes w.r.t. the original version can be summarized as follows:

- We have implemented a web interface (<https://graphsimviz.net>) which allows interactive exploration of our results and hence makes them “actionable” as required by one of the reviewers.
- We have carried out additional analyses using a third disease vocabulary (UMLS CUI), a third version of the graph edit distance, and an additional data source for disease-gene association data (CTD).
- We provide a more detailed discussion of related work and more explicitly explain why our work provides genuinely novel insights that go beyond existing studies.
- We rectified our misleading usage of the term “disease ontology” in the original version of the manuscript to clearly emphasize that network medicine’s (and our) long-term objective to replace phenotype-based by mechanistically grounded disease definitions is a genuinely biomedical rather than a semantic endeavor.

In the following, we provide point-by-point explanations of how we have addressed the reviewers’ comments. In the revised version of the manuscript, all changes are marked in blue.

Reviewer 1

R1.1: The importance of data incompleteness and data bias in gene (or variant) disease associations is not new. For example, Menche et al. have studied this issue comprehensively in <https://www.science.org/doi/10.1126/science.1257601>. The investigative bias in function and disease genomics datasets has been studied in the past using various perspectives, for example, Stoeger et al., “Large-scale investigation of the reasons why potentially important genes are ignored” (PLoS Biology 2018); Haynes et al. “Gene annotation bias impedes biomedical research” (Scientific Reports 2018); Esteban et al., “Information silos distort biomedical research” and “The speed of information propagation in the scientific network distorts biomedical research”; Kustatscher et al., “Understudied proteins: opportunities and challenges for functional proteomics” (Nature Methods 2022) and other papers, some of which are cited by the aforementioned articles.

Our reply: We absolutely agree that there are several studies which have analyzed the impact of various types of data biases related to *genes and proteins* (and, to a lesser extent, also variants). However, in our paper, we focus on the specific data bias which mechanistically

ungrounded disease terms induce in the *disease* part of disease-gene and other disease association data. The fact that this bias exists is well-known in the network medicine field, but our paper is the first one to systematically assess its effect. In order to more clearly distinguish our work from previous studies, we now discuss studies on gene- or protein-induced data bias in the introduction and explicitly state that we are investigating a bias which emerges from the “disease side” of disease association data. As a consequence, we deleted the “Study bias and incompleteness of interactome” subsection from the “Discussion”, since it would have been redundant with the newly added paragraph in the introduction.

R1.2: Considering earlier research already pointing to the issue of data incompleteness and bias, the novelty of the observation put forward in this manuscript is relatively limited. That would not be an issue on its own if the manuscript would quantify the extent of the bias rigorously or develop a novel theory (see my concern #4 [comment R1.7 in our numbering]).

Our reply: We disagree with Reviewer 1’s concern regarding novelty. As explained in our reply to R1.1, our work does *not* focus on the well-studied issue of data incompleteness mentioned by the reviewer. Instead, we focus on the effect of the usage of mechanistically ungrounded disease terms in disease association data, which has never been investigated before. Many members of the network medicine community are aware of this problem but we are the first to quantify and explicitly draw attention to it. Moreover, we would like to stress that the empirical *P*-values provided by our analyses do “quantify the extent of the bias rigorously”. To make these results more accessible and “actionable”, we have now developed the GraphSimViz (“**graph similarity visualizer**”) web tool (<https://graphsimgviz.net>), which allow biomedical researchers to interactively query and visualize the results of our study for user-selected diseases, disease association data types, and disease vocabularies. Moreover, researchers can use the GraphSimQT Python package to carry the GED-based analyses we designed for our study on their own networks.

R1.3: It is unclear to which extent the specific modeling decisions made by the authors influenced the results. For example, the study uses customized versions of the graph edit distance (GED) to compare networks. However, there are many other types of network metrics to compare networks. Why is the selected approach the most appropriate?

Our reply: We added a new subsection “Rationale for using the graph edit distance as a measure of network dissimilarity” to the “Methods”, where we provide a detailed explanation for the choice of GED. In sum, except for equivalent alternatives, GED is the only available graphs distance measure we are aware of which satisfies the following four requirements necessary for our analyses:

1. To allow testing both the global- and the local-scale hypothesis, we need a graph distance measure $d(G_1, G_2)$ which is decomposable into local node distances $d(G_1, G_2, u)$.

2. The local node distances $d(G_1, G_2, u)$ should depend on u 's local neighborhoods in G_1 and G_2 but not on the overall network topologies (otherwise, we would not be testing the local-scale hypothesis when comparing local node distances).
3. Since a node alignment between the compared networks is given (disease and drug terms are aligned between the networks), both the global network distance $d(G_1, G_2)$ and the local distances $d(G_1, G_2, u)$ should be node-identity-aware rather than permutation-invariant.
4. The distances need to be computable in linear time w.r.t. the size of the networks in order to enable our large-scale permutation tests.

In order to further corroborate the robustness of our results w.r.t. the distance measure, we additionally ran all analyses using a third variant of GED with edge edit costs based on normalized edge weights (see newly added passages in “Graph edit distance” subsection of “Methods”). The results – which are shown in Supplementary Figures 9 to 12 – are indeed very similar to the ones obtained for the two variants (rank-based and uniform) we had used for the original submission.

R1.4: Further, the analysis is done using MONDO and ICD-10 codes linked with the OMIM and DisGeNet namespace. How have the authors resolved disease vocabularies (e.g., disease subtypes, uncertain disease matching)?

Our reply: In order to run similarity analyses on different networks constructed from a multitude of databases, we have mapped the disease vocabularies used in the source databases (e.g., OMIM, ORPHA, SNOMED-CT, and MeSH; see Table 1 in the manuscript) to (now) three vocabularies: MONDO, ICD10, and UMLS CUI. MONDO contains mappings between its own disease vocabulary and various other vocabularies and we used this source for the mapping task to MONDO vocabulary. Mapping to UMLS CUI vocabulary was done using the mappings provided in the UMLS Metathesaurus 2022AA full release. Mapping to ICD-10 was done indirectly by mapping the source databases' terms to MONDO, followed by mapping MONDO to ICD-10. This mapping step was also checked manually to make sure the best matching between two vocabularies had been selected and the terms without a good match in the target vocabulary were dropped from the data. When there are specific codes for subtypes, e.g. in MONDO that we have subtypes, they are considered in the mappings. ICD-10 three-char does not include all subtypes, therefore, the cases without disease subtypes are mapped to more general umbrella disease codes. Figure 6 from the manuscript shows the details about the levels of mappability of disease vocabularies underlying our study. Here, we would like to highlight again one of the findings of our study which is the dependence of results on the final (target) vocabularies used for the analyses. The results indicate that integrating different databases with different disease vocabularies can introduce blurriness (by mapping from a finer-grained vocabulary like MONDO to a coarser-grained one like ICD-10).

R1.5: In the analysis of drug-related data, other confounders that could attenuate or emphasize data bias are not discussed. For example, the extent of data bias can depend on therapeutic

area, molecular scaffolds, drug approval date, therapeutic modality, and other biological and biochemical properties, such as chemical classification and MoA, beyond the ATC classification considered in the paper. Are there specific subsets of diseases, drugs, and proteins for which bias is more substantial or weaker? Can the authors disentangle the contributions of various cofounders by defining an appropriate causal model and using partial correlation?

Our reply: We now added a “Limitations” subsection to the discussion, where we discuss cofounders beyond phenotype-based disease definitions such as those mentioned by the reviewer which might explain the results. However, we would like to stress that the results we obtained for our comparisons of diseasomes were very similar independently of the type of disease association data used to construct the diseasomes. Since the phenotype-based disease definitions affect all diseasomes while other biases such as the drug-related biases mentioned by the reviewer or incomplete disease-gene association affect only specific diseasomes, this is strong (but of course not conclusive) evidence that the observed results can indeed mainly be attributed to mechanistically inadequate disease definitions.

We would also like to point out that, already for the original manuscript, we had tried to identify “specific subsets of diseases, drugs, and proteins for which bias is more substantial or weaker” but no clear patterns emerged. This is explicitly mentioned in the last paragraph of the “Results of local-scale analyses” subsection in “Results”. For the revised version of our manuscript, we carried out additional analyses where, for each comparison of two disease-disease networks G_1 and G_2 , we fit a linear model $p \sim \beta_0 + \beta_1 \cdot d_1 + \beta_2 \cdot d_2$, where p is the vector of local empirical P -values (one P -value per disease) and d_1 and d_2 are the vectors of the diseases’ node degrees in G_1 and G_2 . We then carried out slope tests to assess whether the coefficients β_1 and β_2 significantly differ from 0 – i.e., whether the local empirical P -values linearly depend on the degrees of the diseases in the compared diseasomes.

The results are shown in Figure 1 in this response letter. Again, we could not identify a clear and interpretable pattern. Although the P -values of the slope tests were significant in many cases, the fact that the direction of the linear dependencies varied across many of the tests makes it challenging to interpret the results. For instance, in the MONDO-based comparison between symptom- and variant-based diseasomes using rank-based edit costs, the empirical P -values significantly increase with increasing node degrees in the symptom-based diseasome, whereas we observe a significant decrease in the corresponding UMLS-based comparison.

Finally, we would like to clarify that we did not use the ATC *classification* system in our study. ATC has 5 levels, where the 5th level is the chemical compound to which DrugBank assigns an ID. While we used these IDs as drug identifiers for our drugomes and drug-disease networks, we did not make any use of the ATC hierarchy.

A) Uniform edge edit cost

B) Rank-based edge edit cost

C) Weight-based edge edit cost

Figure 1. Results of slope tests for linear dependency of local empirical P -values on node degrees in diseasomes. In all heatmaps, the cell (i, j) visualizes the result of the slope test for coefficient β_i in the linear regression model obtained for the comparison of the diseasomes G_i and G_j . While the cell's color encodes the P -value of the slope test with null hypothesis $\beta_i = 0$, the “+” or “-” annotations show the sign of the coefficient β_i .

R1.6: Findings of this study imply that uncritical use of DisGeNET or OMIM databases, which rely on phenotype-based disease definitions, is problematic. How do those findings generalize to other prominent sources of gene-disease association data available in the biomedical community?

Our reply: To assess whether our findings also generalize to other sources of gene-disease association data, we generated a new gene-based disease-disease network using the gene-disease association data from the CTD database and ran the similarity analyses against drug-based, symptom-based, and variant-based disease-disease networks (using MONDO disease IDs).

Figure 2. Comparison of local empirical P -values for GED-based network similarity tests using, respectively, DisGeNET and OMIM (blue curves) or CTD (orange curves) as disease-gene association data sources. (A) Results of analyses using uniform edit cost. (B) Results of analyses using rank-based edit cost. (C) Results of analyses using weight-based edit cost.

As shown in Figure 2 in this response letter, the distributions of the obtained local empirical P -values are very similar to those obtained for the analyses based on DisGeNET and OMIM reported in our manuscript. In particular, we again observe P -values close to 1 for a substantial fraction of diseases, indicating that it is indeed also problematic to uncritically use disease association data from the CTD database. Moreover, we would like to emphasize that we selected DisGeNET as one of the main sources of gene-disease association data for our analyses because it is a very comprehensive source of gene-disease data which includes data from other sources such as UniProt, PsyGeNET, Orphanet, the CGI, ClinGen, and more.

R1.7: My biggest concern with the study is that the key results are not actionable, in the sense that the study does not offer a statistically-grounded strategy to a) quantify the bias in a new study that could help researchers decide what BEV/close-up analyses are appropriate vs. not OR b) mitigate the bias identified. The study writes that “disease association databases should hence be used with care.” However, it is unclear what this means and what concrete statistical/network science/information theoretic technique the study provides to help others use the databases “with care.” Rigorous theoretical or statistical grounding or definitions of these concepts is lacking.

Our reply: In order to make our results more “actionable”, we now developed the GraphSimViz web tool (<https://graphsimgviz.net>), which researchers can use to interactively explore our results and, based on our findings, decide whether or not to use a BEV approach for their studies. Assume, for instance, that a researcher is working on uncovering disease mechanisms for, say, osteoporosis (OP, MONDO ID: 0005298) and wants to get some prior intuition about whether or not starting the analysis with OP-associated genes obtained from DisGeNET or OMIM is likely to yield meaningful results. This researcher could use GraphSimViz to have a look at the local empirical P -values for OP obtained for the permutation tests involving gene-based diseases. The researcher would then discover that most of these P -values are non-significant, which highlights the need for close-up studies to disentangle OP disease mechanisms.

To enable “quantify[ing] the bias in a new study” where researchers would like to test other networks based on other data sources, we provide the GraphSimQT (“graph similarity quantification tool”) Python package, which is freely available on GitHub (<https://github.com/repotrial/graphsimqt>). While we had already developed GraphSimQT for the original submission, we now mention it more prominently in the introduction.

Nonetheless, we would also like to stress that the main objective of our study is not to generate actionable knowledge but to draw attention to a problem which – although widely acknowledged – is very often neglected in practice. Every year, a huge number of studies that aim at uncovering molecular mechanisms of complex diseases are published where the authors start their analyses with disease association data obtained from public data sources without even discussing the possibility that the umbrella disease terms used by these data sources might well distort their results. If, upon reading our paper, some authors of such studies step back to consciously ponder the relevance of this problem for their use case instead of simply using

public disease association data out of convenience, we have reached our main objective. In the revised version of this paper, we now explicitly state this in the introduction.

Reviewer 2

R2.1: I would aim to summarize and interpret the observations a bit more towards their clinical relevance or the way the disease association data is generated. Eg for which diseases is the local neighborhood pairwise more similar across different diseasesomes? Do they belong to specific groups of diseases? Could differences between diseases reflect different degree of biases, number of disease genes, degree of the disease node etc?

Our reply: As mentioned in our reply to R1.5, we had tried to identify specific groups of diseases for which the local neighborhoods are pairwise more/less similar already for the first version of our manuscript but did not observe any clear patterns (and had explicitly mentioned this in the last paragraph of the “Results of local-scale analyses” subsection in “Results”). For the revision, we now additionally investigated whether the obtained local empirical P -values can be explained by the node degrees in the diseasesomes but again did not obtain clear results (see reply to R1.5 and Figure 1 in this response letter for details).

R2.2: Overall enough detail is given to understand the methodology. However, It would be helpful to explain in a few words the way the local- and global-scale hypotheses are tested already a bit earlier in the manuscript (ie in the Introduction where the hypothesis are explained or in the beginning of the Results section) to avoid too much jumping back and forth between Methods and other sections for the reader.

Our reply: In the paragraph following the introduction of the two hypotheses, we now explain that we “compared the distributions of local and global GEDs to GED distributions obtained for randomized counterparts of the compared networks.”

R2.3: The manuscript is well written but there are a few typos. Eg in line 560 there is a missing space before the P.

Our reply: We fixed the mentioned typo and checked the entire manuscript for possible other typos.

Reviewer 3

R3.1: Background information on the scope of diseases that are defined by phenotype-based definitions and that also have known molecular mechanisms would have strengthened the premise of this analysis.

Our reply: As detailed in our reply to comment R3.3 below, we are not primarily concerned with diseases that are defined based on phenotypes and also have known molecular mechanisms but rather with those phenotypically defined diseases for which the molecular mechanisms are unknown. This misunderstanding arose because, in the original version of the paper, we often

used the term “disease ontology”, although “disease vocabulary” would have been more correct. In the revised version of the paper, we now consistently use the term “vocabulary”. See our reply to R3.3 below for more details on this point.

R3.2: This statement, in the Introduction, is unsupported by literature. (line 139) “The biases current disease definitions introduce in large-scale disease association databases such as OMIM and DisGeNET do not even out and such databases should hence be used with care in all fields of data-centric biomedicine.”

Our reply: This statement is not supported by literature because it is the main finding of our study. We now mark this more clearly by starting the sentence as follows: “The main finding of this study is hence that the biases [...]”

R3.3: The rationale for the analysis is based on unsupported assumptions: that phenotype-based definitions should and could be replaced by molecular-mechanism based definitions; which assumes that all phenotype-defined diseases have an known underlying molecular mechanism.

Our reply: We are very grateful for this comment because it allows us to clarify an important potential misunderstanding of our study’s overall objective. The misunderstanding emerged due to our imprecise usage of the term “disease ontology” in the original version of the manuscript. Network medicine’s (and our) long-term programme to replace phenotype-based definitions by mechanism-based definitions is not the semantic endeavor to mechanistically re-define semantic relationships between existing disease terms. We do not want to define a new mechanistic disease ontology on top of existing phenotype-based disease terms. As correctly remarked by Reviewer 3, this is impossible as long as the molecular mechanisms are unknown, which is the case for most diseases defined in terms of phenotypes. Rather, the objective is to uncover the currently unknown molecular disease mechanisms and then dissect umbrella diseases such as “Alzheimer’s disease” or “coronary artery disease” into endotypes which are clearly characterized at a molecular level. That is, network medicine really aims at a mechanism-based disease vocabulary rather than a mechanism-based disease ontology. Compiling a mechanism-based disease vocabulary is a genuinely biomedical rather than a semantic endeavor. In the revised version of the manuscript, we now explicitly state this in the introduction.

R3.4: The study states that they are comparing two ontologies, however, that is not the case. The article compares the mondo ontology with ICD-10, a clinical vocabulary developed for morbidity and mortality code. Mondo is an ontology that was built from combining other disease ontologies along with clinical vocabularies, including ICD-10. ICD-10, is a vocabulary, does not include the semantic relationships between disease classes. The choice of these vocabularies may have biased the results, as the two sources are not distinct.

Our reply: As mentioned in our reply to R3.3, we have replaced the term “disease ontology” by the term “disease vocabulary” throughout the manuscript. We use this term also for MONDO,

because we did not utilize MONDO's semantic layers for our study – that is, we in fact used MONDO as a disease vocabulary. To mitigate the possibility that results might have been biased due to non-distinctness of ICD-10 and MONDO, we now carried out the analyses also with UMLS CUIs as node IDs (to the best of our knowledge, UMLS CUI is distinct from ICD-10). The obtained results were very similar to the results obtained for MONDO (also see reply to R3.5 below).

W.r.t. the choice of MONDO and ICD-10 as disease vocabularies for our study, we would like to emphasize that we did not choose the vocabularies *ad hoc*: MONDO was chosen because, for MONDO, we obtained the highest overall mappability from the vocabularies used by the different data sources underlying our study. ICD-10 was chosen because the comorbidity data from the Estonian Biobank originally uses this vocabulary and mapping it to another vocabulary would have introduced substantial noise in our comorbidity networks (see reply to R3.5 below).

R3.5: The analysis based on only two datasets (mondo and icd-10) limits the conclusions of the study. For some of the analysis only one ontology/vocabulary was utilized. See: (line 183) Note that analyses involving comorbidity data were carried out only in ICD-10 and the comparison between target- and indication-based drugomes only in MONDO namespace.

Our reply: For the revised version of the manuscript, we ran the analyses also on networks where IDs from the UMLS CUI vocabulary were used as node IDs and added the results in the manuscript and the supplement (see updated Figures 4, 5, 6 and updated Supplementary Figures 1 to 5 and 7 to 12). The results are very well aligned with the results for MONDO vocabulary, showing that our previous conclusions still hold.

The reasoning why, for some of the analyses, only one vocabulary was utilized is explained in the “Data integration” part of the “Methods” section – now in more detail than in the original version of the manuscript. For the drugomes comparisons, we only used the MONDO vocabulary because, in these networks, nodes are drugs rather than diseases and so using different disease vocabularies leaves the nodes of the networks unchanged. In the indication-based drugomes, the choice of the disease vocabulary can change the edges of the networks, but, in practice, we observed that the differences are small. Target-based drugomes are not affected at all by the choice of the disease ontology. In order to not overload the manuscript with very similar results, we hence decided to only use MONDO for the comparison of drugomes.

The comorbidity data was obtained from the Estonian Biobank, which uses originally ICD-10 codes. In order to carry out analyses involving comorbidity data in MONDO or UMLS CUI vocabulary, we hence would have had to map the comorbidity data from a coarser-grained (ICD-10) to a finer-grained disease vocabulary (MONDO and UMLS CUI). Although this is possible from a technical point of view (we could have mapped a comorbidity link between two ICD-10 codes to a fully connected bipartite graph of the corresponding MONDO or UMLS CUI IDs), it would have introduced a lot of noise in the obtained comorbidity networks. Since we

were afraid that this noise might overshadow all other effects, we hence decided to carry out analyses involving comorbidity data only in ICD-10 vocabulary.

R3.6: The literature support for the statements, below, are not provided in the manuscript.

(lines-87- 93): “Let D be disease association data of some data type T commonly used by BEV approaches (e.g., disease-gene associations). Further assume that D contains entries D (d1) and D (d2) for two diseases d1 and d2 that share an unknown molecular disease mechanism. Then this shared mechanism should lead to similarities between D (d1) and D (d2), given that D indeed contains useful information about disease mechanisms.” “For instance, we would expect that the diseases d1 and d2 have similar profiles of disease-associated genes, that they exhibit high comorbidity, that they lead to similar symptoms, and that they can be treated by similar drugs.”

Our reply: We have added a reference to Langhauser *et al.* (2018) [DOI: [10.1038/s41540-017-0039-7](https://doi.org/10.1038/s41540-017-0039-7)], where a similar thought is developed. Langhauser *et al.* hypothesized that a joint mechanism related to cGMP signaling is mechanistically involved in a disease cluster containing, among others, ischemic stroke as well as some neurological disorders. Langhauser *et al.* then argued that, if this common cGMP-related mechanism indeed exists, it should manifest in shared disease genes, symptoms, and comorbidities. This way, they could test their hypothesis by comparing associated genes, symptoms, and comorbidities for the diseases in their disease cluster.

R3.7: For most of the figures, the font size is too small for readability, and several figures are missing axis labels.

Our reply: We rearranged the figures and increased the font size. We again checked the entire manuscript and ensured that all axes are labeled. Note that, in order not to crowd the figures with redundant information, we deliberately omitted x- or y-axis labels if they could be shared for entire rows or columns of a figure containing several subplots. If required by the journal editors, we will of course provide figures with redundant labels should the paper be accepted.

R3.8: In some areas of the manuscript, such as the "Discordant disease ontologies", statements are not supported. Nor are the vocabularies referenced. The statement that these vocabularies are discordant is misleading. Figure 6 (Levels of completeness of disease ontology mappings underlying this article) incorrectly labels these vocabularies as ontologies, does not recognize that they have been built for different purposes and thus it is reasonable that they may have non-overlapping disease terms, for example OMIM defines Mendelian genetic phenotypes, MESH includes terms from PubMed, and orphanet is a vocabulary of European rare disease. The UMLS_CUI represents the National Library of Medicine's Unified Medical Language System, which maps disease concepts between SNOMED, ICD, MESH and OMIM, among other vocabularies. This figure has not included ICD-10 as one of the Source Ontologies.

Our reply: We fully agree that our statements about “discordant disease ontologies” in the original version of the manuscript were misleading. In fact, we explicitly acknowledge in the

“Discordantly used disease vocabularies” subsection of the Discussion (renamed from “Discordant disease ontologies” in the original submission) that “[t]he vocabularies have different degrees of granularity, and are generated in different ways and for different purposes”, and we agree that it is thus reasonable that they may have non-overlapping disease terms.

The point we would like to make here is not that different disease vocabularies *are* discordant but that they are *used* discordantly by databases containing disease association data. This is now also reflected in the updated name of the subsection. For instance, both DrugCentral and CTD contain drug-indication data, but while DrugCentral used SNOMED CT to denote the indications (diseases), CTD uses MeSH terms. In the context of network medicine applications, we hence very often have to map data to a joint target vocabulary if we want to jointly leverage the disease-association from various data sources (we now explicitly state this in the “Discordantly used disease vocabularies” subsection). This almost always leads to information loss due to unmappable terms.

In our study, we tested for three different target vocabularies: ICD-10, MONDO, and now also UMLS CUI. In Figure 7, we show the mappability of the source vocabularies used in primary data sources to these target vocabularies. ICD-10 does not appear as source vocabulary in Figure 7 because the only data originally using ICD-10 vocabulary is the comorbidity data from the Estonian Biobank, which we did not map to other disease vocabularies for the reasons detailed in our reply to R3.5.

We also added missing references for all mentioned disease vocabularies (at their first appearances in the manuscript).

R3.9: The statement that ‘Disease names are variable and non-standardized’ (line 328) - is not an accurate statement, disease names are standardized and follow community guidelines. This statement misrepresents the history of disease nomenclature.

Our reply: We agree and removed this sentence from the updated manuscript. Instead, we now write in the “Mechanistically inadequate disease vocabularies” subsection of the discussion: “Since causal molecular disease mechanisms are often unknown, disease names often do not denote such mechanisms but rather reflect the person who coined the disease term (e.g. ‘Alzheimer’s disease’), areas in the body that are affected (e.g. ‘kidney stones’) or symptoms of the disease (e.g. ‘irritable bowel syndrome’).”

R3.10: The presentation of the data figures makes it difficult to view and interpret the outcome of the analysis.

Our reply: To allow interactive and user-friendly visualization of the results of our analyses, we now developed the GraphSimViz web tool (<https://graphsimviz.net>). With GraphSimViz, readers of our paper can easily zoom-in on the results for the diseases, networks, and disease association data types they are particularly interested in (see reply to R1.7). Moreover, we have increased the font sizes in the figures (see reply to comment R3.7).

Reviewers' Comments:

Reviewer #1:

Remarks to the Author:

I thank the authors for addressing all of my critical comments, including my primary concern regarding the outcomes of this study. The manuscript has been considerably revised to provide a statistically-grounded strategy as well as a web tool that other researchers can use to quantify the bias in a new study and decide what BEV vs. close-up analyses are appropriate.

Reviewer #3:

None